

# OH airglow observations with two identical spectrometers: benefits of increased data homogeneity in the identification of the 11-year solar cycle-, QBO-induced and other variations

Carsten Schmidt[1], Lisa Küchelbacher[1], Sabine Wüst[1], Michael Bittner[1,2]

[1]German Remote Sensing Data Center (DFD), German Aerospace Center (DLR), Oberpfaffenhofen, 82234, Germany
[2]Institute of Physics, University of Augsburg (UNA), Augsburg, 86159, Germany

*Correspondence to*: Carsten Schmidt (carsten.schmidt@dlr.de)

**Abstract.** Hydroxyl (OH) radical airglow observations have been performed at the environmental research station 'Schneefernerhaus' (UFS, 47.42° N, 10.98° E) since October 2008, and with continuous operation since July 2009. The

instrumental setup relies on the parallel operation of two identical GRIPS (Ground-based Infrared P-branch Spectrometer) in order to achieve maximum completeness and homogeneity.

After the first decade of observations the acquired time series are evaluated with respect to the main influences on data quality and comparability to those on other sites. Data quality is essentially limited by gaps impacting the completeness. While technical failures are largely excluded by the setup, gaps caused by adverse meteorological conditions can systematically

influence estimates of the annual mean. The overall sampling density is high, with nightly mean temperatures obtained for 3382 of 4018 nights of observation (84 %), but the average coverage changes throughout the year. This can bias the annual mean up to 0.8 K if not properly accounted for.

Sensitivity studies performed with the two identical instruments and their retrieval show that the comparability between the observations is influenced by the annual and semi-annual cycle as well as the choice of Einstein-A-coefficients, which

influence the estimate of the annual cycle's amplitude.

A strong 11-year solar signal of 5.9 ±0.6 K / 100 sfu is identified in the data. The OH temperatures follow the F10.7cm value with a time lag of 90 ±65 days. However, the precise value depends on details of the analysis. The highest correlation ($R^2$=0.91) is achieved for yearly mean OH temperatures averaged around 4th February and the F10.7cm solar flux leading ahead with 110 days. A prominent two-year oscillation is identified between 2011 and 2015. This signal is linked to the quasi-biennial

oscillation (QBO) leading to a temperature reduction of approximately 1 K during QBO westward phases in 2011, 2013, 2015 and a respective 1 K increase in 2012 and 2014 during QBO eastward phases. The amplitude of the semi-annual cycle shows a similar behavior with the decade's minimum amplitudes (~2.5-3K) retrieved for 2011, 2013 and 2015 and maximum amplitudes observed in 2012 and 2014 (~4 K). The signal appears to disappear after 2016 when the solar flux approaches its next minimum. Although it appears as a rather strict 24-month periodicity between 2011 and 2015, spectral analyses show a

more or less continuous oscillation with a period of approximately 21 months over the entire time span, which can be interpreted as the result of a non-linear interaction of the QBO (28 months) with the annual cycle (12 months).



# 1 Introduction

At least since the study of Roble and Dickinson (1989) it is known that the middle and upper atmosphere can provide important

insights into long-term changes of atmospheric temperatures and dynamics due to an increase in the $CO_2$ mixing ratio. But the middle atmosphere (10–100 km), especially the mesosphere lower thermosphere (MLT) region, poses several observational challenges. The number of high-quality observations spanning a period of more than one or two decades is limited to a few time series worldwide. At the same time the MLT is believed to be a transition region between different atmospheric regimes. Although temperatures of both the mesosphere and the thermosphere are decreasing, the mesospheric cooling leads to a

contraction of the atmosphere accompanied by a shift of the vertical temperature structure. Warmer parts of the thermosphere are supposed to descend to lower heights, creating a region at MLT heights where the decrease in temperature is supposedly compensated by the descending warmer parts (e.g., Laštovička et al. (2008), Laštovička et al. (2006) and references therein). Therefore, long-term studies of MLT temperatures are of special importance for a better understanding of future developments of middle atmosphere dynamics. Only a few measurement techniques grant access to this region well above the stratopause

and below the ionosphere. Unlike lidar or radar systems, passive remote sensing of natural airglow emissions originating in heights between 80 km to 100 km provides a convenient way of exploring the MLT. Systematic ground-based observations of the rotational-vibrational transitions of the hydroxyl (OH) molecule and the derivation of rotational temperatures from these emissions date back to the 1960ies (see e.g., Semenov, (2000)). While Semenov (2000) had to combine data from several sites in his analysis, long time series of a single site are meanwhile available for a few sites, e.g., at the Kjell Henriksen Observatory

(KHO), 78.2° N, 16.0° E, since 1980; at the University of Wuppertal (WUP), 51.3° N, 7.2° E, since 1980, at the Australian Antarctic Station Davis (DAV), 68° S, 78° E, since 1995; El Leoncito (LEO), 31.80° S, 69.29° W since 1997; Maimaga (MAI), 63.04° N, 129.51° E, since 1999, Zvenigrorod (ZVE), 55.69° N, 36.77° E since 2000 and several other places (see Holmen et al. (2014), Kalicinsky et al. (2018), French et al. (2020), Reisin and Scheer (2002), Ammosov et al. (2014), Dalin et al. (2020)). Especially, a potential lack of homogeneity and completeness poses a threat to long-term OH observations. An instrument

failure for example will initially cause a gap in the time series reducing the completeness. If it cannot be repaired the instrument will be replaced with a new instrument. Even if both instruments are well-calibrated, the successor will most likely differ in terms of sampling rate, throughput and other details impacting the homogeneity of the time series. The further the start of observations dates back the more inhomogeneities and large gaps occur (see e.g., Kalicinsky et al. (2016), Holmen et al. (2014)).

As was pointed out by Laštovička and Jelínek (2019) not only data quality but also analysis methodology and natural variability are sometimes underestimated in trend calculations which can lead to controversial results. In their review and later updates Beig et al. (2003) address solar variability, more precisely the 11-year Schwabe cycle, as the major natural forcing impacting long-term trend estimates. Among other aspects Laštovička and Jelínek (2019) discuss the multiple regression method,





frequently applied in order to decompose a time series into its underlying components such as a bias, cyclic components, long-term trend and noise. They point out, that the more sophisticated/ complex the analysis method is, the more sensitive it will be to data errors. Ultimately, the issues of instrumentation, data quality, natural variability and methodology led to the foundation of the network for the detection of mesospheric change (NDMC; https://www.wdc.dlr.de/ndmc/) in 2007 with its mission to foster the exchange of existing know-how, and the coordinated development of improved observations, analysis techniques and modeling.

Shortly thereafter, the prototype of a new instrument was set into operation at the environmental research station 'Schneefernerhaus' (UFS) in central Europe (47.42° N, 10.98° E) in 2008 (see Schmidt et al. (2013), Bittner et al. (2010)). Based on the previous experience of other research groups in the NDMC, later routine observations were planned with two identical instruments (being each other's backup) from the very beginning. In this study, the first eleven years (07/2009 until 06/2020) covering the decade 2010-2019 and solar cycle 24 are analyzed with respect to data quality and natural variability. It specifically addresses the question whether the direct comparison of the two instruments' yearlong field observations confirms previous estimates of precision, accuracy and homogeneity which before could only be based on laboratory measurements, theoretical calculations and empirical 'best practice' efforts.

Fig. 1 gives an overview of the generic steps involved in acquiring a long-term time series, including the respective specific steps concerning the OH airglow observations discussed in this study. Once the parameter to be observed has been identified (*here*: OH rotational temperatures, representing kinetic MLT temperatures), an appropriate instrument has to be chosen or developed (*here:* a grating spectrograph for the bright OH emissions between 1.5µm and 1.6µm). Then, a certain retrieval needs to be applied to the calibrated observational data (*here*: interpretation of the $P_1(2)$-, $P_1(3)$- and $P_1(4)$-rotational lines of the P-branch of the OH(3-1) vibrational transition). This will result in the desired parameter (OH rotational temperature). In order to get a time series of daily, monthly or yearly mean values for further interpretation several averaging procedures will usually be applied to the initial data, which may have a temporal resolution of only a few seconds or minutes (10 s, 15 s or 30 s seconds in case of this study). This study focuses on the procedures shown with solid arrows/boxes in Figure 1 corresponding to retrieval, averaging and their consequences for later interpretation and comparison to other observations. It concludes with an analysis of the features observed during the first decade of observations.





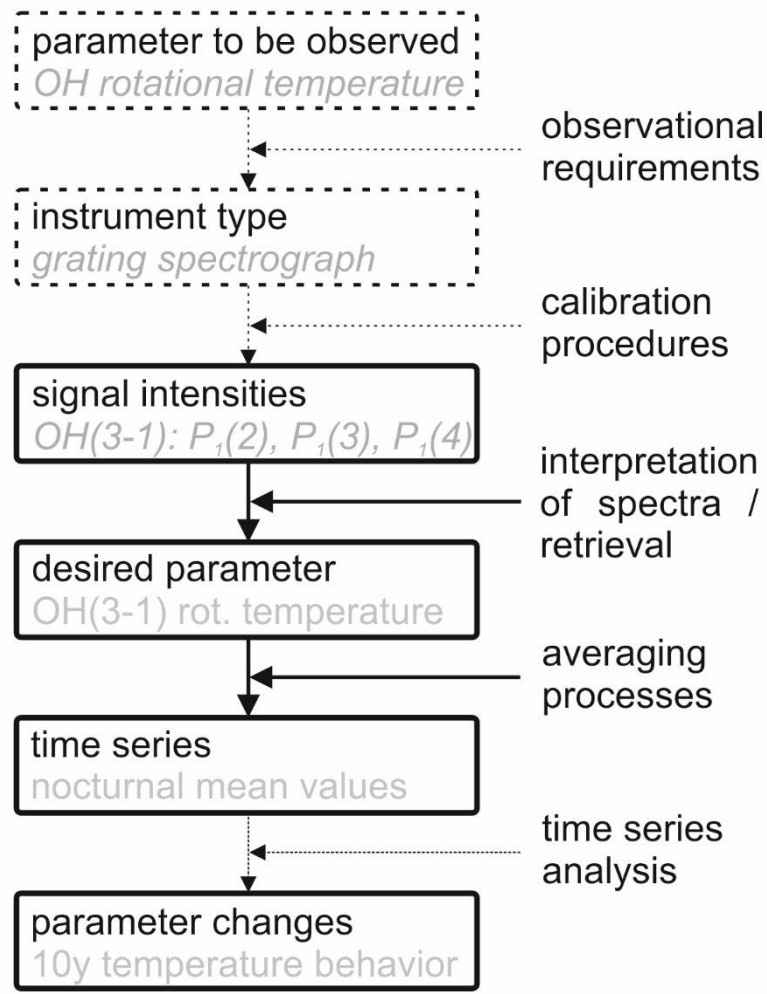

**Figure 1: Typical steps involved in designing an operational observation program and respective analyses. The left column shows the general stages (black) and their implementation in this study of OH-airglow (gray). Necessary steps to get from one stage to the next one are indicated in the right column. Each step can significantly influence later results. This study focuses on the procedures shown with solid arrows/boxes (that is: retrieval, averaging and their consequences for later interpretation) and it concludes with an analysis of the features observed during the first decade of observations.**



The structure of the paper is organized as follows: general aspects of the instruments and temperature retrieval are presented in section 2. The main part in section 3 deals with the remaining differences between the individual instruments and challenges to be met in combining the individual time series. A special emphasis is laid on the non-random appearance of gaps (concerning both nightly and annual mean values). Finally, the time series is used for estimating the solar forcing impact and studying a

quasi-biennial oscillation (QBO) signal, which seem to be the strongest natural influences impacting the validity of potential future trend estimates. The conclusions to be drawn for future analyses are summarized in section 4.

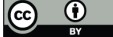



## 2 Instrumentation and data reduction methods

### 2.1 GRIPS

The GRound-based Infrared P-branch Spectrometers (GRIPS) are designed to resolve the $P_1$-lines of the OH(3-1)-rotational
vibrational transition for the derivation of the respective rotational temperature (that is: a resolving power of $\lambda/\Delta\lambda$ of ~500 or
a full width at half maximum (FWHM) of 3.1 nm at 1550 nm). All three instruments used in this study are equipped with an
ANDOR (Oxford Instruments) spectrograph (Shamrock 163, F/#=3.6) and a thermoelectrically cooled InGaAs photodiode
array (iDus InGaAs 1.7) with 512 pixels.

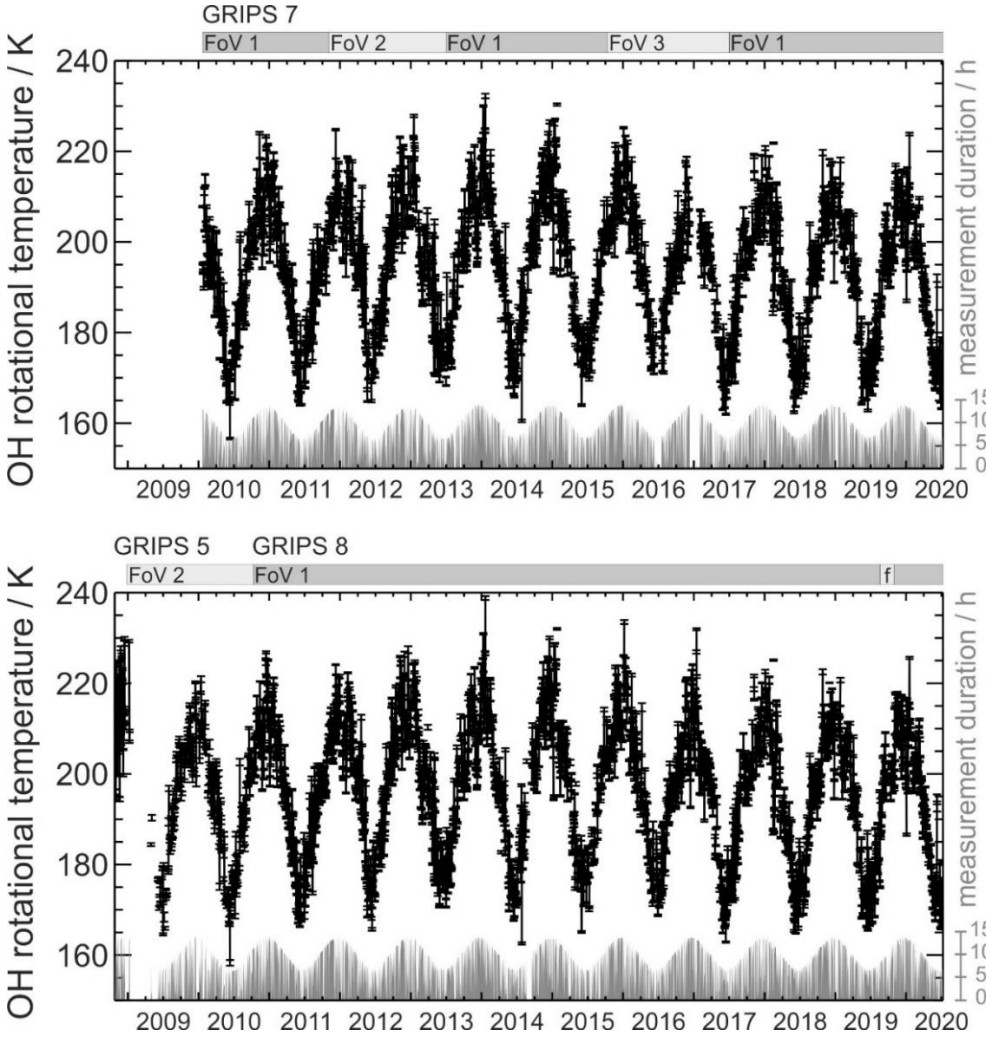

**Figure 2: Nightly mean OH temperatures obtained at the UFS "Schneefernerhaus" since October 2008 (black) and respective
measurement duration (gray). The upper panel shows data obtained with GRIPS 7, the lower panel shows data obtained with GRIPS
5 and GRIPS 8. The grey bars above each panel indicate time spans of different fields of view (FoV), "f" marks a technical failure,
which resulted in flawed values of GRIPS 8 during a couple of weeks in 2019; see Table A1 for a comprehensive overview of
instrument performance.**



More details concerning setup and data reduction schemes of the instruments are given by Schmidt et al. (2013). A few differences concern the fact, that the instruments at UFS are operated with an oblique field of view (FoV) and varying temporal resolution. In their standard setup the instruments are pointing southward at a zenith angle of 45°. Due to their entrance angle of approximately 15.5°x15.5° this corresponds to a FoV of a few hundred square kilometers at 86 km-87 km, the height of the emitting layer (see e.g., Wüst et al. (2020, 2017), von Savigny (2012), Baker and Stair (1988)). The observational filter

introduced by spatial averaging due to the FoV-size has been analyzed by Wüst et al. (2016) in their study of gravity wave potential energy. The systematic differences introduced into the long-term time series by the varying FoV are discussed in section 3.1 of this study. Fig. 2 includes horizontal bars indicating which FoV was used at the respective time (FoV 1: 180° azimuth, 45° zenith angle; FoV 2: 180°/61°; FoV 3: 124°/57°). The standard temporal resolution of all GRIPS is 15 s, but GRIPS 7 outperforms most of the other instruments and is therefore operated at 10 s resolution, while in the early setup

GRIPS 5 suffered from a minor misalignment and its resolution was reduced to 30 s until it was replaced with GRIPS 8. Thus, depending on season and temporal resolution the number of spectra obtained each night ranges between 800 and 5100.

Due to their large entrance angle the precise calibration of the instruments is a challenging task, as the entire field of view needs to be homogeneously illuminated. Originally, a quartz halogen lamp (Bentham Instruments Ltd. CL2) and a calibrated transmission diffuser (SphereOptics Ltd. SG 3210) were used. In a joint effort with the *Physikalisch-Technische Bundesanstalt*

(PTB, Germany's national institute of standards) ongoing efforts of improving the calibration chain began in 2015. Meanwhile, a substantial amount of results has been obtained, and an adequate presentation thereof is beyond the scope of this study. But it should be noted that the GRIPS instruments outperform the original calibration equipment (halogen lamps) concerning long-term stability. Since a calibration offset between GRIPS 7 and GRIPS 8 of 1.83 K was only precisely identified after years of parallel observation, this offset is kept to emphasize that the instruments are independently calibrated. It is of course corrected

for, once the data of the individual instruments are combined.

Observations started on 25th October 2008 with a prototype setup of GRIPS 5. Although the instrumental setup was not changed since mid-December 2008, autonomous observations frequently failed until June 2009. Therefore, general data quality is considered sufficiently high for long-term analyses since July 2009. Measurements using GRIPS 7 have complemented these observations since 20th January 2010. GRIPS 5 was replaced with GRIPS 8 on 2nd September 2010. While the layout of

GRIPS 7 was changed for a couple of sensitivity studies, GRIPS 8 remains unchanged since the beginning - thereby providing a stable reference. A good overview of the instruments' performance is provided by their effective observation time displayed at the bottom of the graphs (gray area) in Fig. 2.

## 2.2 Raw data reduction

The first step in determining the temperature of the OH layer involves applying the physical relationship between the emission

intensities $I_{v',J',i}$ and the OH molecules' rotational temperature $T_{rot}$. It was already given by Meinel (1950):



$$I_{(v',J',i \to v'',J'',i)} = N_{v'} \cdot A_{(v',J',i \to v'',J'',i)} \cdot \frac{2(2J'+1)}{Q_{v',T_{rot}}} \cdot \exp\left(\frac{-F_{v',J',i}}{k_B T_{rot}}\right). \quad (2.1)$$

Here, $v'$ ($v''$) and $J'$ ($J''$) denote the vibrational level and the quantum number of angular momentum of the upper (lower) state of the molecule with i being the doublet branch under investigation; the transition observed with GRIPS is OH(3-1)-$P_1$ ($v'$=3, $v''$=1, i=1 / electronic state $X^2\Pi_{3/2}$). A, $F_{v',J',i}$ and $k_B$ denote physical constants: A, the Einstein coefficient of spontaneous emission, F, the term value of the rotational level and $k_B$ is the Boltzmann constant. In principle, it is possible to derive $T_{rot}$ from a single emission line described by equation (2.1), given a high-quality absolute calibration is available for the observed emission intensities $I_{v',J',i}$. However, the required accuracy and precision of the calibration are hard to achieve and in addition the population number $N_{v'}$ as well as the state sum $Q_{v',Trot}$ are not readily accessible but constant for lines of a single branch. Therefore, the first three lines $P_1(2)$, $P_1(3)$, and $P_1(4)$ of the $X^2\Pi_{3/2}$ state are observed and a set of equations is formulated and linearized by taking the logarithm. It takes the shape of an inverse problem: $\underline{y} = \underline{G} \cdot \underline{x}$, with observations (line intensities) summarized in vector $\underline{y}$ (dimension: *1 x n*), unknown properties in parameter vector $\underline{x}$ (*m x 1*) and the physical relationship between both summarized in matrix $\underline{G}$ (*n x m*); n refers to the number of lines observed (here: n=3) and m to the number of unknown quantities (here: m=2, for $T_{rot}$ and the ratio $N_{v'}/Q_{v',Trot}$). This inverse problem is solved for $T_{rot}$ by applying the least squares method, leading to:

$$\underline{x} = \left(\underline{G}^T \underline{G}\right)^{-1} \underline{G}^T \underline{y}. \quad (2.2)$$

Although applying the least squares method is considered straightforward in this case, it implicitly attributes more weight to emission lines which are further apart in the spectrum (here: $P_1(2)$ and $P_1(4)$). Retrieving correct estimates for the intensities of these two lines from the observed spectra is therefore crucial. Since the $P_1$-lines partially overlap with the $P_2$-lines (of the $X^2\Pi_{1/2}$ state), line intensities are estimated by only considering the intensities at the line centers. A correction is still required as $P_1(4)$ is close to $R_1(6)$ of the OH(4-2) transition (1543.22 nm vs. 1543.02 nm, approximately one order of magnitude smaller than the spectral resolution of GRIPS). Their overlap is temperature dependent and an iterative correction algorithm was provided by Lange (1982) and re-evaluated by Schmidt (2016). The results of their studies (which may be difficult to access) can be summarized as follows: at 150 K the correction amounts to -1 K and is exponentially increasing for increasing temperatures with -3 K at 200 K and -13 K at 250 K. Alternative approaches indicate a precision of ±0.5 K of this correction, with the overall performance decreasing beyond 230 K and failing to converge beyond 283 K (see also Fig. 3).

## 2.3 Einstein-A-coefficients in the retrieval

According to Eq. (2.1) the relationship between emission intensities and rotational temperature requires knowledge of several physical constants. But unlike the precisely known Boltzmann constant, $k_B$, several sets of constants have been published for both Einstein-A-coefficients of spontaneous emission and rotational term values F. The Einstein-A-coefficients of the hydroxyl transitions have previously been identified to be of great importance for certain OH-vibrational branches (e.g., Noll et al.



(2020), Liu et al. (2015)). In their analysis of the OH(6-2) transition French et al. (2000) found temperatures can differ by up to 13 K depending on the set of Einstein-A-coefficients.

The influence of the Einstein-A-coefficients in the GRIPS retrieval is further investigated in more than 20,000 observed spectra of varying temperature and signal-to-noise ratios of GRIPS 8. Fig. 3 shows the temperatures retrieved from these observation
data based on eight different sets of Einstein coefficients, including the coldest (warmest) summer (winter) periods observed between 2010 and 2020. The figure displays the differences to temperatures retrieved with coefficients of Mies (1974) – the standard coefficients still used in the GRIPS retrieval. An important property concerns the fact that the Mies (1974) coefficients lead to higher temperatures than the more recent coefficients of Brooke et al. (2016), van der Loo and Groenenboom (2008) or HITRAN 2016 (High Resolution Transmission molecular spectroscopic database, Gordon et al. (2017), Goldman et al.
(1998)). Moreover, the differences depend on the absolute temperatures and thus seasons: in winter (at ~230 K) the differences are twice as large as in summer (at ~160 K). This influences further comparisons, as Mies (1974) coefficients will lead to systematically higher annual amplitudes than those from Brooke et al. (2016).

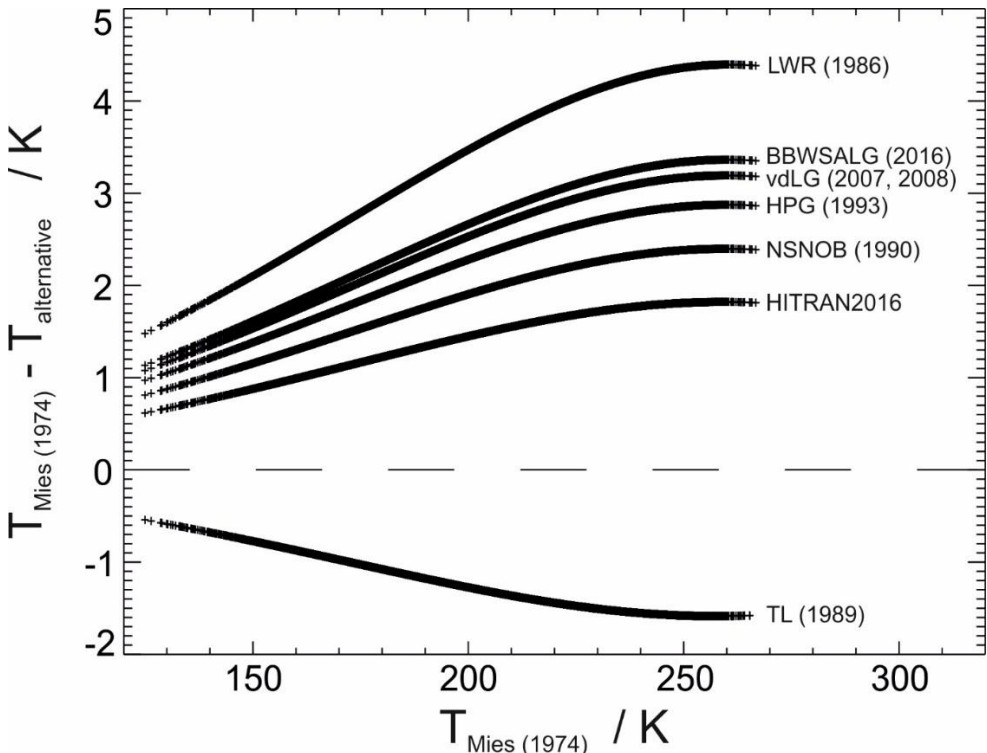

Figure 3: **Differences between rotational temperatures, if different sets of Einstein A coefficients are applied during the (otherwise
identical) retrieval. The temperatures are obtained from more than 20,000 spectra obtained with GRIPS 8 during cold (summer) and warm (winter) conditions. The temperatures of this study have been calculated with Einstein coefficients published by Mies (1974) unless otherwise noted. BBWSALG - Brooke et al. (2016), HPG - Holtzclaw et al. (1993), LWR - Langhoff et al. (1986), NSNOB – Nelson et al. (1990), TL – Turnbull and Lowe (1989), vdLG - van der Loo and Groenenboom (2007, 2008), HITRAN – High Resolution Transmission molecular spectroscopic database, Gordon et al. (2017), Goldman et al. (1998). The linear relationship
is evident from 130 to 230 K, but a deviation from it occurs at higher temperatures. This is due to the correction algorithm discussed in section 2.2.**


However, the temperatures based on the different coefficients can be exactly converted from one set to the other, as these retrievals are not influenced by the signal-to-noise ratio of the observations: each "line" in Fig. 3 consists of more than 20,000 individual data points. Actually, the differences exhibit a linear relationship, the curvature at higher temperatures is due to the decreasing accuracy of the correction algorithm outlined in section 2.2.

Unless otherwise noted, the temperatures from the GRIPS instruments are still based on the Einstein-A-coefficients from Mies (1974) and rotational term values taken from Krassovsky et al. (1962). The latter are less often discussed, because the influence of different sets of term values is smaller by one order of magnitude. Concerning the GRIPS retrieval only the values of Kvifte (1959) may lead to differences of up to ±0.4 K. Other sets of term values lead to temperatures which are systematically lower but no more than -0.1 K. In general, the respective differences exhibit a smaller temperature dependence and are to some extent influenced by the signal-to-noise ratio, see also Fig. A1 in the appendices. In contrast to the Einstein-A-coefficients employed, the influence of the term values on the retrieval is negligible and not discussed any further here.

## 3 Results and discussion

### 3.1 Combining data from different instruments / sites

Occasionally, the data completeness at a measurement site or the sampling rate of a sole instrument is insufficient for an intended analysis of longer term variability. Thus, time series from different instruments or sites have to be combined. Semenov (2000) was among the first to combine OH temperature time series from several sites (even continents) acquired by different operators with different instruments and retrievals. Despite many corrections applied before comparing these data this can easily compromise the homogeneity of the time series under investigation. Since the pioneering work of Semenov many lessons have been learned and better corrections are usually applied today.

Offermann et al. (2010) combined data from two sites (separated by ~500 km) acquired with identical instruments and retrievals. They compared each data set individually to satellite observations with TIMED-SABER (Thermosphere Ionosphere Mesosphere Energetics Dynamics, Sounding of the Atmosphere using Broadband Emission Radiometry). The satellite-based instrument thus served as a transfer standard. They found that despite some minor differences these data sets agree within their uncertainty estimates. However, their instruments (GRIPS 1 and 2) are predecessors of the instruments used in our study and the higher precision of GRIPS 5,7 and 8 in combination with more than 10 years of parallel observations allows a more detailed investigation of the challenges to be met in combining OH temperature time series. In addition, comparisons between ground- and satellite-based instruments involve certain challenges of their own, e.g., the observed volumina rarely match exactly, leading to the so-called miss-time and miss-distance errors (see e.g., Reisin and Scheer (2017), Wüst et al (2016), Wendt et al. (2013), French and Mulligan (2010) and references therein).



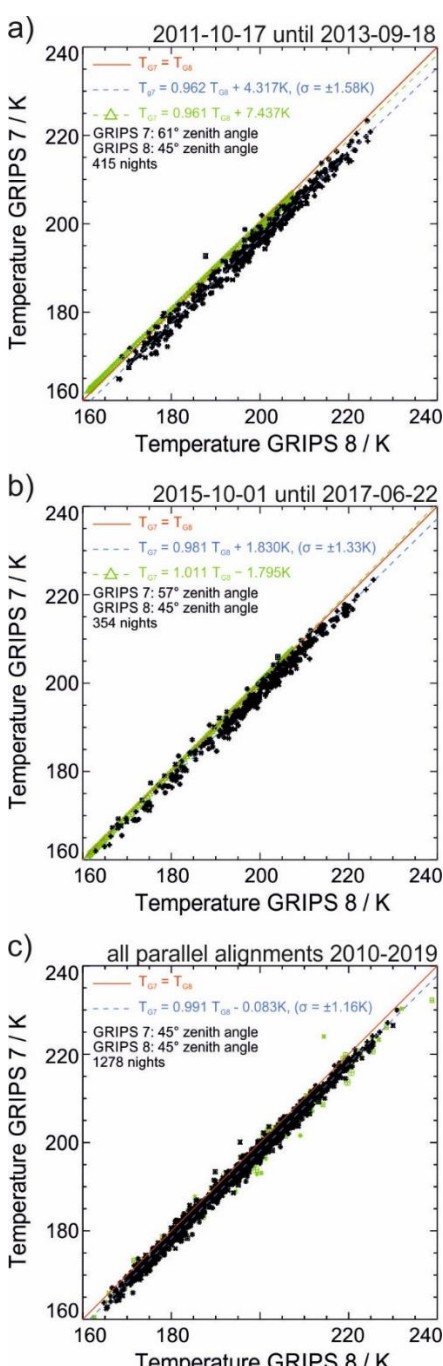

**Figure 4: Scatterplots of GRIPS 7 vs. GRIPS 8 derived rotational temperatures for different years (bisecting lines: solid red, fit to the observed data: dashed blue, MSISE00 climatology values: dashed green triangles), GRIPS 8 stays unchanged with zenith angle of 45° and azimuth of 180° throughout all years: a) both instruments pointing southwards, but GRIPS 7 has a larger zenith angle of 61°, b) both zenith and azimuth angles of GRIPS 7 are changed to 57° and 124°, respectively, c) all parallel alignments between 2010 and 2019; green data points in c) are excluded later due to more restrictive criteria than in the original processing scheme (see also Figure 6). Details are discussed in the text.**



**Table 1: Systematic differences between the temperatures obtained with GRIPS 7 and GRIPS 8. Changing only the effective latitude of the GRIPS 7 FoV (Oct 2011 – Sep 2013) had the most significant influence on the differences. The repeated parallel alignments**
**show no systematic drift between the instruments. Note: Due to repairs affecting the optics of GRIPS 8 this comparison ends in August 2019. Including data extending beyond the time frame of this study the systematic differences currently remain at a level of 1.77 K ±1.05 K (0.994 $T_{G8}$ - 0.607 K).**

| Observation period | FoV of GRIPS 7 zenith / azimuth | Mean difference $T_{G8}$-$T_{G7}$ (±1-sigma) | $T_{G7}$ = f($T_{G8}$) | Number of nights |
|---|---|---|---|---|
| 2010-09-02 until 2011-10-16 | parallel | 1.98 K ±1.34 K | 0.995 $T_{G8}$ − 1.09K | 256 |
| 2011-10-17 until 2013-09-18 | 61° / 180° | 3.10 K ±1.58 K | 0.962 $T_{G8}$ + 4.32K | 415 |
| 2013-09-18 until 2015-09-30 | parallel | 1.71 K ±1.07 K | 0.987 $T_{G8}$ + 0.80K | 492 |
| 2015-10-01 until 2017-06-22 | 57° / 124° | 2.00 K ±1.33 K | 0.981 $T_{G8}$ + 1.83K | 354 |
| 2017-06-22 until 2019-08-02 | parallel | 1.86 K ±1.13 K | 0.991 $T_{G8}$ + 0.15K | 530 |
| all parallel alignments | 45° / 180° | 1.83 K ±1.16 K | 0.991 $T_{G8}$ - 0.08K | 1278 |

Since October 2010 the instruments GRIPS 7 and GRIPS 8 have been operated in a parallel setup with identical FoV. Only
twice the viewing direction of GRIPS 7 was changed, when Wüst et al. (2016) studied the impact of the observational filter due to the size of FoV in gravity wave observations. Therefore, the zenith angle of GRIPS 7 was increased slightly from October 2011 until September 2013, so that it just did not overlap with the FoV of GRIPS 8, while keeping the 180° (tward) azimuth angle. Subset a) of Fig. 4 shows the respective scatter plot of GRIPS 7 temperatures versus GRIPS 8 temperatures (black) with 415 nights covering almost two seasonal cycles. The inclination of the linear fit (dashed blue) to the data is 0.962
(±0.005) and thus deviates slightly from 1.0 (solid red). This inclination is in remarkable agreement with the respective inclination derived from the MSISE-climatology for both latitudes (0.961 ±0.001), although the absolute temperatures taken from MSISE are lower (dashed green triangles). Subset b) shows the time period from October 2015 until June 2017 when the setup of GRIPS 7 was changed again, only this time it not only pointed at a different latitude but also at a different longitude further away from the GRIPS 8 FoV. Based on 354 nights the observed temperature dependence (0.981 ±0.005) does not agree
with the one expected from MSISE-climatology (1.011 ±0.001) this time. The remaining parallel alignments summarized in subset c) amount to 1278 nights showing a temperature dependence of 0.991 ±0.002 (black dots with error bars), which is





close to the expected 1.0. The nights shown with green symbols have been excluded from analysis due to more restrictive

quality criteria defined below (see Section 3.4 and Fig. 6). Both the mean difference of 1.83 K and the 1-sigma scatter of

1.15 K are significantly smaller, when the FoV agree. The individual parallel alignments (shown in Fig. A2 in the appendices)

indicate that the temperature dependence is reproducible with individual values of 0.995, 0.987 and 0.991.

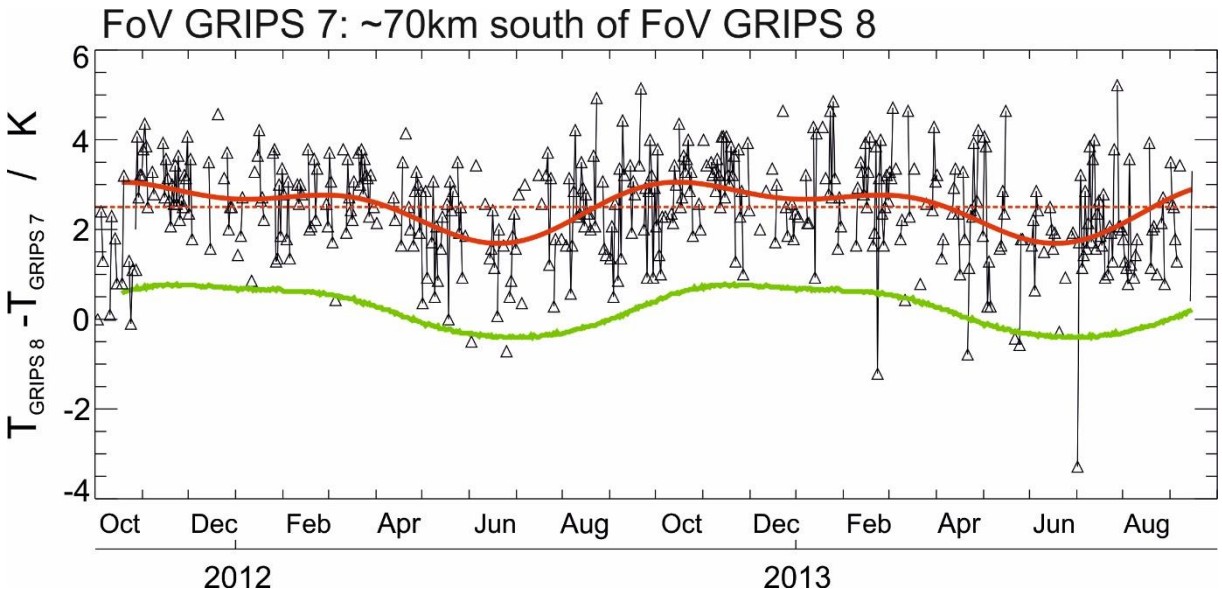

**Figure 5: Observed differences between the nocturnal temperatures (black), when the FoV of GRIPS 7 was changed from a zenith**
**angle of 45° to 61° showing the same data as Fig. 4a). The green line shows the differences expected based on MSISE00 climatological**
**means. The solid red line shows the result of a Harmonic Analysis, reproducing the annual and semi-annual oscillation. The dashed**
**red line shows the (bias-corrected) mean difference.**

Fig. 5 shows the temperature differences calculated by using the two instruments for the time period, when the GRIPS 7 FoV

was changed, so it just did not overlap with the GRIPS 8 FoV anymore. The mean difference and variability (1-sigma) increases

by approximately 0.7 K (from 1.8 K to 2.5 K, Note: due to the distribution of gaps the observed difference noted in Table 1 is

biased and appears to be higher). Furthermore, the difference shows a seasonal behavior: it is larger from August to April (up

to +1.2 K in October/November) and smaller during summer from May to July (down to -0.2 K in May). The green line shows

the differences, which are expected when the MSISE00 climatological means for the respective positions (at 87 km and local

midnight) are adopted. Fitting the annual and semi-annual components to the data with a harmonic analysis almost perfectly

reproduces the course of the MSISE climatology. Obviously, the seasonal cycle can explain the different slopes observed in

the scatter plot in Fig. 4a). Unfortunately, technical failures in 2015 and 2016 caused large gaps (see "measurement duration"

in Fig. 2 and Table A1), so no reliable results can be obtained for the second time period, when the FoV was changed.

The temperatures obtained from the two instruments are not drifting apart, which is proven by the repeated parallel alignments.

Yet, the temperatures begin to show systematic differences once small changes between the setups are introduced (see Table





1). These systematic differences will further increase if data are compared to data from other observing sites. Since these differences are caused by latitudinal differences and their respective seasonal cycles, the temperature difference between two observing sites can hardly be approximated by a simple fixed offset; even a linear relationship as shown in Fig. 4 may improperly represent the differences between two sites. Although Offermann et al. (2010) performed a thorough analysis before combining their data from Wuppertal (51° N, 7° E) and Hohenpeißenberg (48° N, 11° E), the larger variability in their data

probably prevented the identification of a seasonal dependence beyond their estimated offset of 0.8 K to 1.2 K between the two sites.

**3.2 Uncertainty of annual and nightly mean values**

Since long-term time series of OH temperatures are usually composed of nightly mean values, the quality of the entire time series depends on both quality and quantity of the individual nightly mean values. The quantity is easily determined in case of

the UFS time series: once the data from all instruments are combined (with systematic biases shown in Fig. 4 and 5 removed) there are on average 287 valid nightly mean values available per year (~78.5%).

The situation is more difficult concerning the quality of the nightly mean. It is usually expressed via precision $\Delta T$ (addressing random scatter) and accuracy, approximated by the bias $T_{Bias}$ to a reference (accounting for systematic errors); in the following quantities with "$\Delta$" refer to quantities, which increase the random scatter and those with subscript "B" refer to those quantities,

which can have a systematical influence. $\Delta T$ is mainly composed of four contributions:

$$\Delta T = \Delta T_{stat} + \Delta T_{samp} + \Delta T_{meth} + \Delta T_{cal}. \tag{3.1}$$

Here, $\Delta T_{stat}$ is a statistical uncertainty, mainly driven by the number of individual temperature values contributing to the nightly mean value. $\Delta T_{samp}$ is the uncertainty due to the actual sampling over the course of the night (e.g., clouds can significantly

influence the value of $\Delta T_{samp}$ by obstructing the FoV). Sometimes it is also referred to as 'geophysical noise' caused by incomplete averaging of atmospheric waves, especially tides, correlating the individual observations (e.g., Beig et al. (2003)). $\Delta T_{meth}$ is the uncertainty due to intrinsic properties of the methods applied in the retrieval, it is often negligible. An example of $\Delta T_{meth}$ coming into play is the exponential relationship between OH line intensities and rotational temperatures: to a small extent higher temperatures will therefore have higher uncertainties (see Eq. 2.1). $\Delta T_{cal}$ represents the precision of the calibration

chain from the field calibration of the instrument via secondary standards to some national reference source. These reference sources are radiance or irradiance standards, determining their contribution to $\Delta T$ involves sophisticated methods, which is beyond the scope of this study.

Similarly, $T_{Bias}$ depends on several contributions:



$$T_{Bias} = T_{B,cal} + T_{B,samp} + T_{B,meth} + T_{B,repr}.$$

(3.2)


Here, $T_{B,cal}$ is the calibration offset of the instrument; $T_{B,samp}$ is a potential bias introduced by the actual observational coverage over the course of the night; $T_{B,meth}$ is a bias introduced by the methods applied in the retrieval. A well-known contribution to $T_{B,meth}$ is the mandatory application of Einstein-A-coefficients in the retrieval, with different sets yielding different results (see Fig. 3). $T_{B,repr}$ addresses the question, to which extent OH rotational temperatures are representative for the kinetic temperature

of the atmosphere.

The contributions of $\Delta T_{meth}$, $\Delta T_{cal}$ and $T_{B,cal}$, $T_{B,meth}$, $T_{B,repr}$ are assumed to be (more or less) constant over time. The remaining $\Delta T_{stat}$, $\Delta T_{samp}$ and $T_{B,samp}$ can vary between individual nights. It is especially difficult to estimate $\Delta T_{samp}$ and $T_{B,samp}$. Commonly, the effective observation time during a given night is taken as a criterion to identify nights, when contributions of $\Delta T_{samp}$ and $T_{B,samp}$ are negligible. But there is no common agreement concerning how long this effective observation time needs to be. For

GRIPS a value of two hours was empirically derived early in 2009 (Schmidt et al., 2013). Pautet et al. (2014) use a value of at least four hours, Reisin and Scheer (2002) use a value of at least 3.6 hours – sometimes required to be distributed over an actual observing time of six hours. Both point out that these are typical values referring to the majority of their observations but that ultimately the averaging needs to meet the requirements of the intended analysis (Dominique Pautet, Esteban Reisin, Jürgen Scheer, personal communication). Other researchers apply different approaches; e.g., Bittner et al. (2002) use a lower

limit of at least one hour of observation time, with a scientist carefully approving each night in addition. A similar statement was brought forward by Lowe: "*depending on your site one hour of observation time might be sufficient, if sampled around local midnight*" (Bob Lowe †, personal communication). The latter statement points at the primary issue of sun-synchronous tides being able to systematically shift the airglow temperatures depending on the local time of observation. Scheer et al. (1994) were among the first to compensate for tides biasing their means due to incomplete observational coverage by using data from

neighboring nights.

By investigating the large data set comprising ten years of parallel observations with two identical instruments, it is possible to investigate this sampling issue in greater detail. Fig. 6 shows the temperature differences between nightly mean values as a function of the difference in observation time. Each instrument acquired data for at least two hours. Subsets a) and b) show the data during times with identical as well as different FoV. During the majority of nights, the measurement duration does not

differ more than 45 minutes (see overplotted bar graph). The average observation time of GRIPS 8 is longer, because the temporal resolution of GRIPS 8 is 15 s and only 10 s for GRIPS 7. Thus, the signal received by GRIPS 8 stays longer above the detection threshold during adverse conditions.





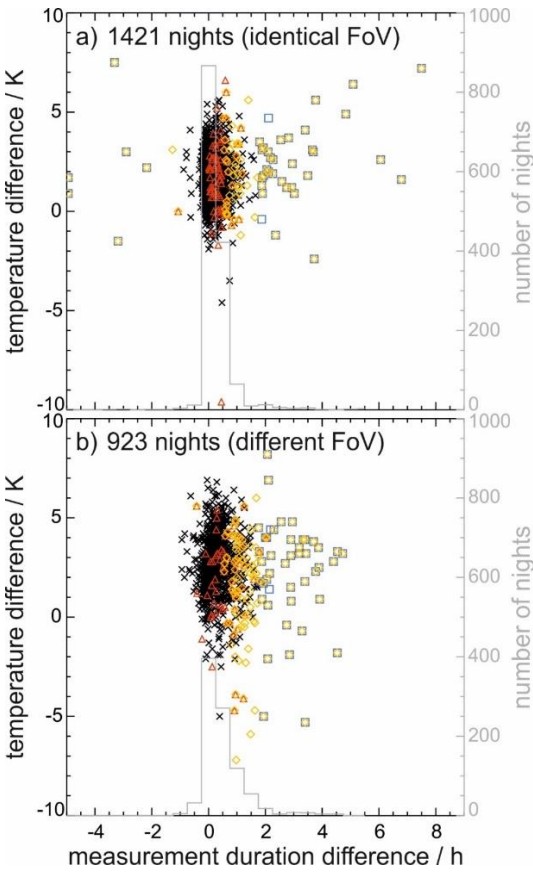

**Figure 6: Temperature difference as a function of the instruments' individual measurement duration shown as observation time**
**GRIPS 8 minus observation time GRIPS 7 for a) matching FoV and b) for separated FoV. Black crosses denote a minimum observation time of at least 3 h, an absolute difference of less than 1.75 h and a relative difference of less than 17.5%; red triangles show nights with observation times of two to three hours; blue squares (yellow diamonds) show nights with observation time differences of more than 1.75 h (17.5%). During the majority of nights, the observation time agrees within 45 minutes.**

Both cases show that about 3 % to 5 % of the nights are outside of the majority of the distribution. The selection criteria can

be adjusted by excluding nights with an absolute observation time difference of more than 1.75 h (blue squares), a relative

observation time difference of more than 17.5% (yellow diamonds) or an observation time of less than 3 h (red triangles). The

remaining black crosses denote the data which have been used in the scatter plots in Fig. 4. Although only 50 nights appear to

be outliers in Fig. 6a) more than 140 nights are excluded from further analysis by adding this last criterion. Obviously, the

remaining differences still range from roughly -1 K to +4.5 K which is larger than their calibration offset (ca. 1.83 K) combined

with the typical measurement uncertainty $\Delta T$. Note that $\Delta T$ is solely based on $\Delta T_{stat}$ in the standard processing scheme (v1.0),

resulting in less than ±1 K for each instrument (in 99 % of the nights), see Schmidt et al. (2013) for a detailed discussion of

this scheme. But the observed scatter in Fig. 6 indicates that the true value of $\Delta T$ is often closer to ±2 K (or ±2.75 K/$\sqrt{2}$).



Traditionally, ΔT is almost entirely based on $\Delta T_{stat}$ neglecting contributions from the other sources (given the selection criteria

outlined above are applied). This approach yields reasonable values assuming that the individual temperature values are uncorrelated and exhibit a random scatter, which is larger than the systematic variations caused by atmospheric waves. Then, ΔT can simply be estimated by σ/√n, with the standard deviation, σ, and the number of individual data points, n. However, the instruments' superior temporal resolution makes $\Delta T_{stat}$ become fairly small; GRIPS 7 for instance acquires more than 700 spectra within two hours. In addition, statistical approaches usually require the individual data points to be independent. But

atmospheric waves change the individual temperatures in a deterministic way.

Fig. 7a) shows the average course of temperatures during January for the decade 2010 to 2019; subsets b) to d) show three successive nights observed in January 2018. While the first night shown in Fig. 7b) is characterized by almost eight hours observation time, meeting each of the quality criteria mentioned above, the other two nights actually cover sunset to sunrise. Comparing the first night to the January mean, one should expect its respective mean to be too high by 0.7 % or 1.5 K. But if

the other two nights were limited to the observation time of the first night, their retrieved means would actually be too low by 0.7 K (Fig. 7c) or hardly change at all (Fig. 7d). While Fig. 7a) shows the influence of sun synchronous tides, especially the semi-diurnal tide, specific nights often show variability due to gravity waves.

**Table 2: Comparison of the estimates for precision and accuracy of the nights shown in Fig. 7. The actual observing time (length and position within the night) can strongly influence both, the sampling bias $T_{B, samp}$ and the precision ΔT. The last column shows the arithmetic mean of the observation time, indicating whether or not the observations are evenly distributed around local midnight (~23:17 UTC).**

|  | Arithmetic mean (as in Fig. 7) | Classic approach data version 1.0 | New approach data version 1.0A | Observing time | $T_{B, samp}$ | Mean time of observation / UTC |
|---|---|---|---|---|---|---|
| 2018-01-11/12 | 211.6 K | 213.8 K ±0.1 K | 211.8 K ±1.6 K | 7.96 h | 1.93 K | 02:59 |
| 2018-01-12/13 | 216.2 K | 218.6 K ±0.1 K | 218.6 K ±0.1 K | 13.68 h | 0.01 K | 23:49 |
| 2018-01-13/14 | 210.2 K | 212.2 K ±0.1 K | 212.2 K ±0.1 K | 13.68 h | 0.01 K | 23:48 |






Hence, a systematic influence of tides ($T_{Bias}$) exists, but a respective quantitative estimate thereof also needs to include a proper treatment of the related precision $\Delta T$. The influence of atmospheric waves (tides and gravity waves) on both parameters is therefore based on a statistical approach. For each month the 100 best nights observed in this decade serve as a reference (covering the entire night from sunset to sunrise without gaps). The quality of any new night is now judged by investigating

how the retrieved nightly means of the reference nights change, once the actual observation time of the observed night is applied to the reference nights. If the gap pattern systematically shifts the mean temperatures of the reference nights towards lower or higher values, then it is causing a systematic sampling bias $T_{B,samp}$. If the gap pattern only introduces a certain scatter of the nightly means, with some being higher and others being lower than before, then it is only causing an increase in sampling precision $\Delta T_{samp}$. The respective new data processing does not change the initial interpretation of the spectra. Therefore, it is

documented as version 1.0A, based on the version 1.0 retrieval.

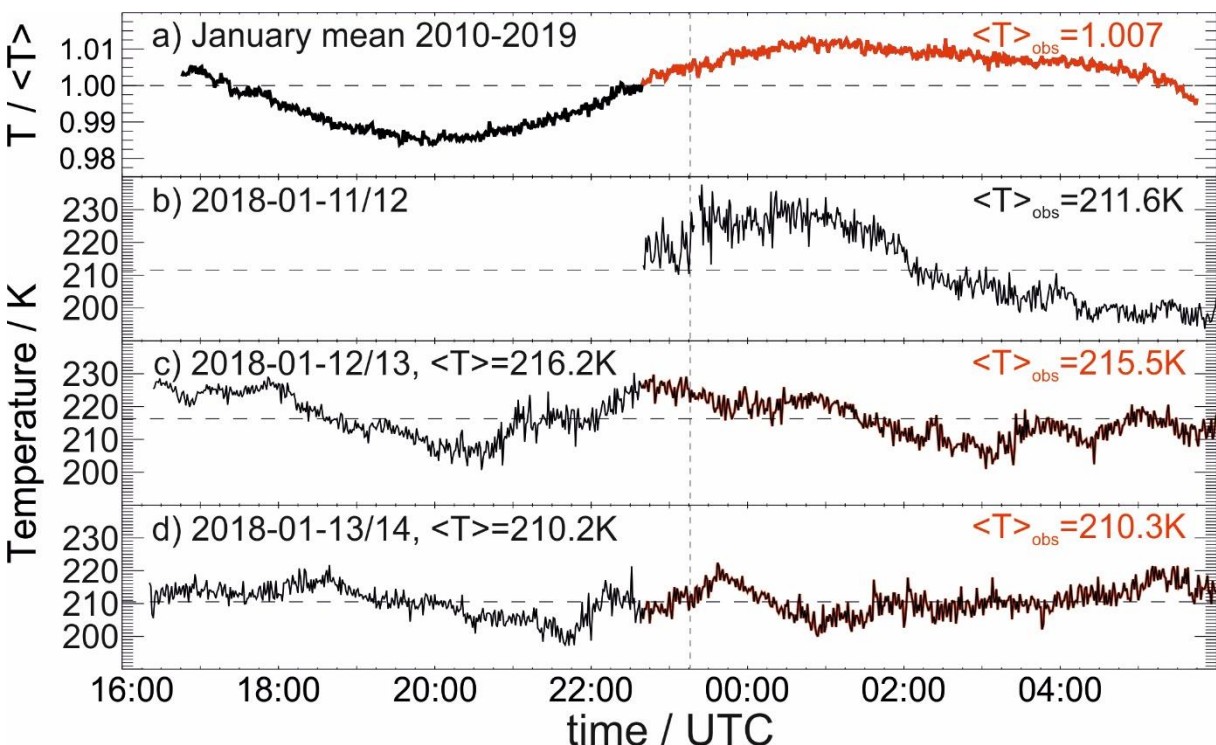

**Figure 7: Improving the estimation of the nightly mean. Subset a) shows the average January night of the decade 2010-2019. The red part highlights the observation time of 11th/12th Jan 2018 shown in subset b). Therefore, the observed mean value <T>obs of**
**211.6 K is expected to be too high by 0.7% (=1.5 K) for this night. The successive nights shown in subset c) and d) indicate, that this statistical relationship does not always apply as the respective sampling bias in these nights would actually be -0.7 K or +0.1 K. See text for further details.**



Table 2 shows the respective changes for the three nights shown in Fig. 7. Despite the long time span of almost eight hours, the limitation to the second half of the night causes high estimates for both $\Delta T_{samp}$ (± 1.6K) and $T_{B,samp}$ (+1.93K) in case of the first night. The better the sampling for a given night is, the smaller $\Delta T_{samp}$ and $T_{B,samp}$ become (see the other two nights). The systematic influence of sun-synchronous tides is the main contributor to $T_{B,samp}$, while $\Delta T_{samp}$ is mainly governed by the random influence of gravity waves and nonmigrating tides. Note: Fig. 7 displays simple arithmetic means (Table 2: column 2) for

easier comparison of these values with the plots. The actual processing involves weighting procedures, both in the former data version (column 3) as well as in the new version (column 4), see Schmidt et al. (2013) for more details.

The reassessment of these uncertainties is expanded to the entire data set. Fig. 8 shows nightly means of the years 2014 and 2018. Red triangles and black dots in subsets a) and b) refer to data retrieved via the former and newer processing, respectively. For the former version, the principal factor in determining the validity of one such nightly mean was the observation time

(minimum of two hours). The improved retrieval now allows for a better assessment of the data quality. In Fig. 8 only those nights with an uncertainty of less than two percent (corresponding to $\Delta T = \pm 4$ K at 200 K) are displayed. In this case, the number of nights available for investigation actually increases from 292 nights to 317 nights in 2014 and from 298 to 319 in 2018. If a more restrictive criterion is applied, for instance only nights with one percent uncertainty are used, then the number of nights will be slightly smaller than before. On average 307 nights per year (84%) qualify for further analysis in this improved data

version 1.0A if an uncertainty of less than two percent is required.

While this improvement in error assessment is important for individual nights, the consequences for the annual means are rather small with a change of only 0.1 K and 0.2 K. This indicates that nights with positive and negative sampling biases occur rather randomly throughout the time series. It is the expected behavior, because incomplete measurement nights are mostly due to bad weather, which on average does not show any systematic occurrence during the night at UFS. Subsets c) and d) of

Fig. 8 show the differences between the two retrievals for both years. Although the differences range from +3 K to -4.5 K they are indeed more or less randomly distributed and thus have little effect on long-term means. However, there are episodes with specific features that should be kept in mind for certain analyses. During January / February 2014 the sampling bias $T_{B,samp}$ is often clearly positive for one night and negative for the next one. During April (August) 2014 $T_{B,samp}$ is systematically high (low). This might be related to tidal activity, but a detailed investigation is beyond the scope of this study. Apparently, several

nightly means lying further away from the majority of the nights are characterized by rather low precision (e.g., low temperatures below 165 K in June/July 2014 or high temperatures above 220 K between October and December 2018). But other 'extreme' events such as the temperature variability in February 2018 ranging from 225 K to 185 K are clearly of atmospheric origin.

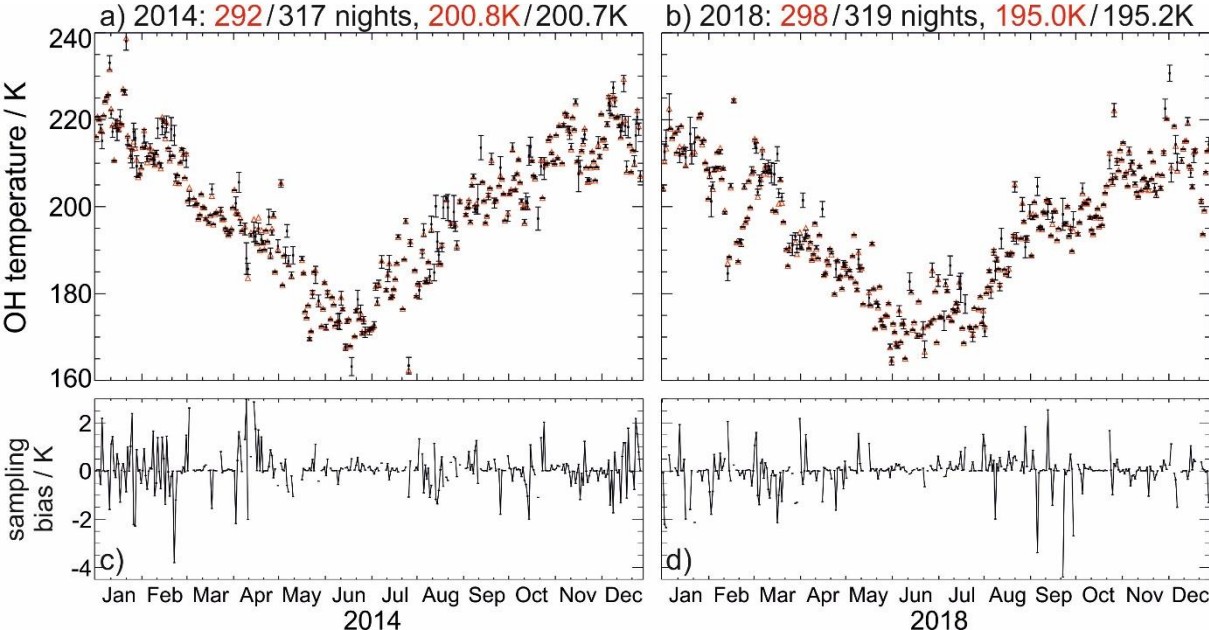

**Figure 8: Comparison of the classical (red triangles) retrieval and the improved retrieval (black dots with error bars) for the years 2014 ( a) & c) ) and 2018 ( b) & d) ). The influence of the retrievals on the annual mean are low, because the number of nights with negative or positive sampling biases are almost equal ( c) & d) ). However, individual nights can be off by up to 4.5 K (September 2018) and there are episodes with short-term deviations (e.g., April and August 2014). See text for further details.**

### 3.3 Annual means and seasonal components

In order to reduce complexity and to remove seasonal features from the data for long-term analyses, annual means are often derived from the time series of nightly means. Similar quality criteria as for nightly mean values apply. Especially, nonuniform sampling during the year can cause a bias of the annual mean. Fig. 9 shows the annual means from 2010 to 2019; each panel represents different data reduction schemes, starting with different Einstein-A-coefficients in panel a). The differences of the respective annual means are almost constantly 2.6 K (with little variance in the second decimal place). If the remaining gaps are filled before calculating the mean, the annual means will be smaller in six out of ten years (see discussion of gap filling below). This systematic bias is largest in 2015 with 0.8 K. A detailed analysis reveals that gaps are often not uniformly distributed throughout the year, but in May and June there are systematically fewer observations due to prevailing higher cloud coverage in the troposphere. This is the time when temperatures approach the annual minimum, (see Fig. 2, 'measurement duration', e.g., in 2011 or 2015). In some years this is compensated by comparable gaps in winter (when temperatures reach their yearly maximum, e.g., in 2019).

Currently, a dozen sites are equipped with an instrument of this GRIPS type. But UFS is the only site with two instruments operated in parallel to avoid gaps due to technical failures. Thus, it's important to investigate to which extent technical failures change the retrieved annual means and if and how these biases can be corrected. The largest gaps occur in the GRIPS 7 data in 2016 (47 nights) and 2017 (42 nights), changing the mean 2017 by 1.4 K (see Fig. 9b and Table A1)). The 2016 mean is





less affected, as both low summer values and high winter values are missing. The individual data agree better, if all gaps are filled with the respective ten-year mean for each missing night (Fig. 9c). However, in 2017 the GRIPS 7 data still show the largest discrepancy to the other time series (0.48 K difference). Since 10-year means (covering high and low temperatures / solar activity conditions) may not be available for every observing site or even introduce a new unknown bias, an independent method for filling the gaps is tested: a harmonic analysis (HA, for further information see, for example, Wüst and Bittner

(2006)) fits the annual and semi-annual component to each year. Annual means are then systematically higher by 0.29 ±0.08 K as the HA does not well reproduce the summer minimum with these settings. In addition to the HA, the Maximum Entropy Method (MEM) can be used in its capacity as linear prediction filter as described by Bittner et al. (2000). The MEM belongs to the autoregressive methods, which regards the current state of a time series as a linear combination of (a finite number of) past states plus a random input. A great benefit of the MEM is, that it can be successfully applied to rather short time series.

Thus, a missing value can be estimated based on only a few preceding values, while the properties of the MEM ensure that the first statistical moments (mean and variance) of the time series are not changed. A comprehensive overview of the benefits and caveats of the MEM is given in Bittner et al. (1994). Although the MEM is not designed for filling large gaps, the annual means (dotted line) agree well with the former method (0.06 K ±0.25 K), because the combination of HA and MEM better reproduces the high-frequent variation of the real data (the largest gaps to be filled comprise 14, 12 and 10 nights, see also

Table A1). Although a terannual component of the HA is often used for the analysis of the OH data (e.g., Ammosov et al. (2014), Bittner et al. (2000)), it has not been used here. The terannual component might better represent the existing data but not correctly estimate the missing data in case of large gaps. Since the vast majority of observation sites in the NDMC are equipped with only one instrument, the possibility to largely reduce the influence of gaps with these methods is regarded as reassuring.

**Figure 9: Annual mean OH temperature at the station UFS: solid red curves with triangles and solid blue curves with squares show the same data in all subsets; a) combined data from all instruments; the application of different Einstein-A-coefficients in the retrieval leads to a shift of the annual mean (red triangles according to Mies (1974) vs. gray diamonds based on Brooke et al. (2016), here: 2.6 K). Interpolating the remaining gaps before calculating the mean shifts the annual means up to -0.8 K in some years (solid blue and dashed black); b) annual means of the individual instruments' data (dashed: GRIPS 7, dotted: GRIPS 5/8, red: same as in a)), the numbers denote the instrument with technical failures during the respective year (see Table A1 for details on large gaps); c) same as b) but missing nights have been replaced with the decadal mean for the respective nights. Apparently, the influence of the gaps has been eliminated to a large degree (blue: same as in a)); d) alternative method of gap filling demonstrated at GRIPS 7 data; gaps have either been filled with a harmonic fit (HA) of the annual and semi-annual component each year (dashed) or with the HA and an additional application of the MEM (dotted); the blue line is identical to a) and c).**



The seasonal components used in treating gaps are also important parameters in studying atmospheric dynamics. Fig. 10 shows amplitudes of the annual and semi-annual components (figure parts a and b) as well as their respective phases (figure parts c and d). Three different approaches were taken: 1) application of Einstein-A-coefficients from Mies (1974) and without interpolation of gaps due to missing nights (solid gray triangles), 2) application of the same coefficients with interpolation of gaps (solid black squares) and 3) application of Einstein-A-coefficients from Brooke et al. (2016) with interpolation of gaps (dashed black diamonds). The differences between these three approaches are largest in the assessment of the annual amplitudes. If gaps are not properly treated, the amplitude of the annual component is mostly smaller, up to 0.6 K in 2013, with slightly higher variability. This is due to the fact that data gaps appear in most cases not at the zero-crossings of the annual cycle. The application of different Einstein-A-coefficients simply shifts the amplitude by approximately 0.4 K. In general, all three approaches show the same variation, which is characterized by (referring to the solid black line) smaller amplitudes (from 16.8 K to 17.2 K) in 2013, 2016 and 2019 and higher amplitudes (from 17.9 K to 18.4 K) in 2014, 2015 and 2018. The semi-annual amplitude varies between 2.3 K and 4.1 K. Einstein coefficients do not influence the results for the semi-annual amplitude significantly (less than 0.1 K). But not properly treating the gaps prior to analysis again increases the variability. Between 2011 and 2015 the semi-annual component resembles the two-year periodicity of the annual mean temperatures shown in Fig. 9, with maximum values of ca. 4.1 K in 2012 and 2014 but minimum values of only 2.4 K to 2.7 K in 2011, 2013 and 2015. After a hiatus in 2016, the two-year periodicity appears to continue (further discussed in section 3.4). Subset c) shows the phases of the annual components. As can be expected from the nature of the annual cycle, the phase is rather constant excepting the year 2016, when the phase shifts by 0.15 rad corresponding to a shift of approximately nine days. The phases of the semi-annual component shown in subset d) exhibit greater variability corresponding to shifts of ±15 days, but no correlation with other parameters is observed.



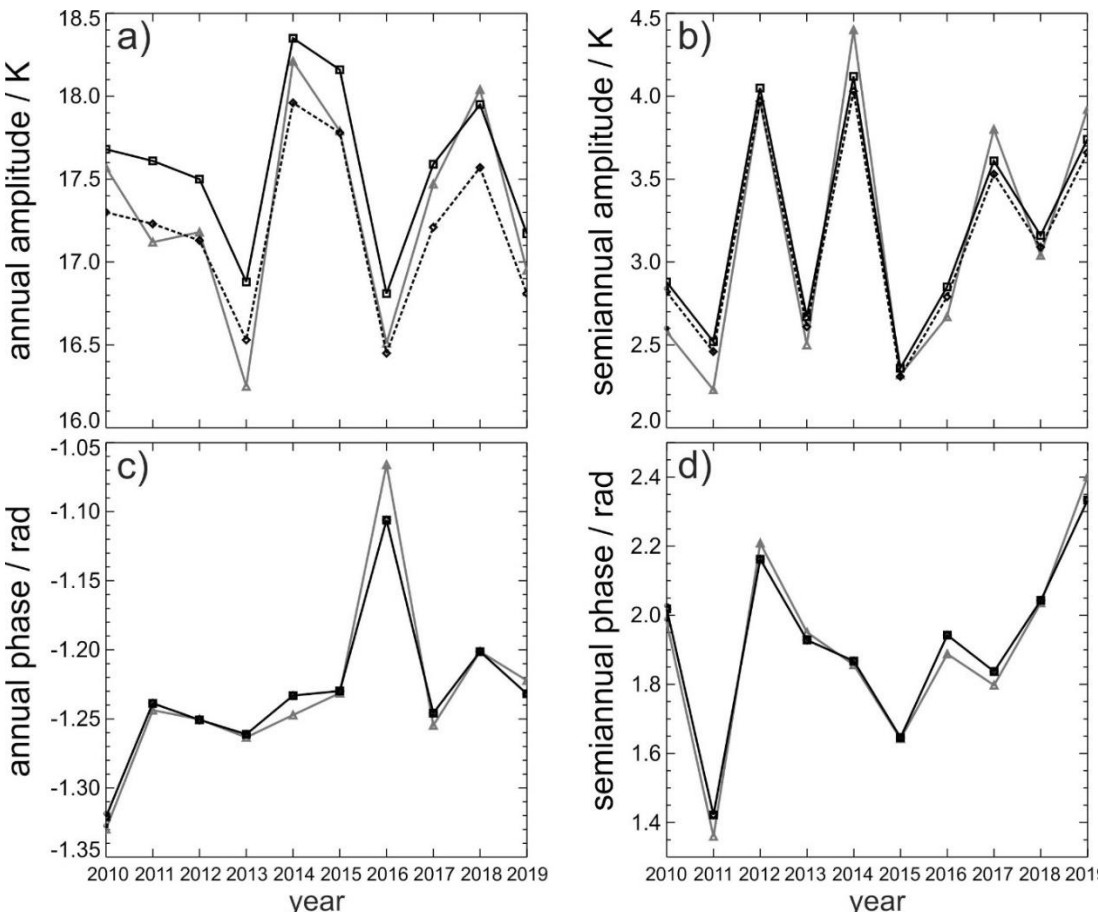

**Figure 10: Amplitudes and phases of the annual and semi-annual component; a) amplitude of the annual component, based on the retrieval with Einstein-A-coefficients taken from Mies (1974) and including gaps (gray triangles) or gaps removed (solid black, squares) and based on retrieval with the Einstein-A-coefficients from Brooke et al. (2016) and gaps removed (dashed black, diamonds); the older coefficients lead to a systematically higher amplitude (~0.4 K) and interpolating the gaps before the calculation can shift the amplitude by up to 0.6 K; b) same as a) but for the semi-annual component; c) the same as a) but for the phases; d) the same as c) but for the semi-annual component. Note: the solid and dashed black curves are almost indistinguishable in c) and d).**

### 3.4 Solar cycle effect and QBO

The correlation between solar activity (widely expressed by the proxy F10.7cm index) and OH temperatures has been studied by many researchers earlier (see Table 3 for a summary) and high correlation values are evident in the analysis of the UFS data sets, which cover the ~11-year solar cycle 24 almost completely as discussed in this section. Fig. 11a) shows yearly means of the OH rotational temperatures, averaged over different temporal intervals. The black line represents the 365-day moving average, effectively eliminating the annual and shorter harmonic components (input data range: 1st July 2009 until 30th June 2020). The colored lines highlight the yearly means based on either the Jan-Dec average (red triangles) or the Jul-Jun average (blue squares). There are two striking features: 1) lower temperatures around 195 K/196 K from the beginning of the time series until mid-2011 and again since 2016 to the end of the time series, 2) a comparatively strong quasi-biannual component





between 2011 and 2015, when temperatures are around 198 K to 200 K. It appears that this quasi-biannual component is also present prior to 2011 and beyond 2015, but with a much weaker amplitude (see also Fig. 12). Fig. 11b) shows the F10.7cm radio flux (extracted from NASA/GSFC's OMNI data set through OMNIWeb: https://omniweb.gsfc.nasa.gov, accessed on 8 Sep 2020) in black and a 365-day moving average in blue. High OH temperatures are correlated with a high solar flux, and, remarkably, the quasi-biannual component seems to be weakening with decreasing OH-temperatures / decreasing solar activity. Fig. 11c) shows the QBO wind data at 30 hPa (from Freie Universität (FU) Berlin: https://www.geo.fu-berlin.de/en/met/ag/strat/produkte/qbo/index.html, accessed on 6 July 2021).

The correlation between OH temperatures and solar flux is strongly influenced by the quasi-biannual component: in 2011 mean OH-temperatures appear colder (-1 K for QBO eastward), while in 2012 they appear warmer (+1 K for QBO westward). Also the solar flux variation shows two distinct maxima in early 2012 and in 2014, both coincide more or less with QBO westward phases. However, the two-year periodicity of the OH temperatures covers a longer time period and roughly resembles the QBO wind variation but only during increased solar activity from 2011 until 2015 (compare red triangles in Fig. 11a) and 11c)).

Studies of the stratospheric wind regime have discussed such a link between QBO and solar cycle for several decades (e.g., Labitzke (1987); see review by Baldwin and Dunkerton (2005) and references therein). According to Labitzke (2005) the Brewer Dobson circulation (BDC) enhances during solar maximum conditions and westward phase of the QBO leading to a warmer polar stratosphere and weaker polar vortex. Accordingly, the QBO westward phases observed in 2011, 2013, 2015 and (out of sequence) 2016 correspond to relatively lower MLT-temperatures. While the OH temperatures increase by approximately 4 K during solar maximum conditions, the QBO then forces a modulation of ±1 K onto this temperature increase. The relationship between QBO and OH-temperatures was first studied by Neumann (1990). Nikolashkin et al. (2001) performed a similar study and found a clear anticorrelation between their OH temperatures and the westward phase of the QBO. However, their data may not have been well suited for this kind of analysis (1201 nights over a period of nine years (with large gaps) sampled at different observing sites used by Neumann (1990) and only 152 nights over a period of six years used by Nikolashkin et al. (2001)). Batista et al. (1994) studied the correlation of the F10.7cm index with OH(9-4) derived temperatures (and other airglow parameters) regarding also the QBO by fitting harmonics to their observation data extending from 1977 to 1986. Their analysis attributed an amplitude of 0.41 K to the harmonic representing the QBO.

A comprehensive overview of the coupling between QBO and mid- to high-latitude mesospheric temperatures was given by Espy et al. (2011) who explicitly dealt with the fact that a clear QBO-signal cannot be found in many OH temperature time series. They propose that the (tropical) QBO modulates the residual circulation from winter to summer pole and thus the mesospheric summer temperatures (which cannot be observed via OH airglow at high-latitude sites). Our observed phase relationship with warmer (colder) summertime MLT temperatures during westward (eastward) winds of the QBO matches the explanation proposed by Espy et al. (2011)). Although a QBO-signal is also present in our summertime data, its amplitude is larger during wintertime: between 2011 and 2015 the year to year change of monthly mean temperatures amounts to 1-4 K in July but 6-9 K in February and 3-8 K in October. Thus, the QBO-related signal in our data appears to mainly originate from variability at the beginning and at the end of Northern hemispheric (NH) winter.



**Table 3: Impact of solar activity on OH airglow temperatures. Only studies published within the last decade have been incorporated. Several studies deal with the same data set but discuss different analysis methods or data periods. Therefore, "location" indicates the underlying time series, while "data period" and "data reduction /analysis methods" give details on how the "solar cycle influence" was estimated. Bold numbers refer to the authors' recapitulatory conclusions.**

| Reference | Location | Solar influence (K/100sfu) | Data period | Time lag | Data reduction/ analysis methods applied |
|---|---|---|---|---|---|
| Holmen et al. (2014) | 78° N, 16° E | **3.6 ± 4.0** | 1983-2013 | - | winter means (Nov-Feb) |
| | | 3.7 ± 0.9 | | - | nightly means |
| | | 4.4 ± 2.4 | | - | monthly means |
| | | 5.1 ± 2.7 | | - | monthly means excl. SSW |
| | | 8.8 ± 0.9 | | 65 days | nightly means |
| | | 6.4 ± 2.3 | | 65 days | monthly means |
| | | ca. 0 | 1983-2001 | - | winter means (Nov-Feb) |
| | | 10.9 | 2001-2011 | - | winter means (Nov-Feb) |
| Ammosov et al. (2014) | 63° N, 130° E | **4.24 ± 1.39** | 1999-2013 | 0 months | monthly averages (Aug – Apr) |
| | | ca. 7 | | 25 months | monthly averages (Aug – Apr) |
| Perminov et al. (2014) | 57° N, 37° E | **3.5 ± 0.8** | 2000-2012 | - | nightly means (multiple regression) |
| Perminov et al. (2018) | | **4.1 ± 0.5** | 2000-2016 | - | yearly means (both: Jan-Dec and Jul-Jun, m. r.) |
| Dalin et al. (2020) | | **3.35 ± 3.22** | 2000-2018 | 0 years | winter (Oct – Mar) |
| French and Klekociuk (2011) | 68° S, 78° E | 3.77 ± 0.69 | 1995-2010 | - | nightly means (multiple regression) |
| | | 3.89 ± 0.69 | | - | monthly means (m. r.) |
| | | **4.79 ± 1.02** | | - | winter means, Mid-Apr - Mid-Sep (m. r.) |
| | | 4.35 ± 0.87 | | - | yearly means (Mar - Oct, m. r.) |
| | | 4.63 ± 0.63 | | 160 days | monthly means |
| | | 4.20 ± 0.66 | | 0 | monthly means |
| French et al. (2020) | | **4.30 ± 1.02** | 1995-2018 | - | winter means, Mid-Apr - Mid-Sep (m. r.) |
| Offermann et al. (2010) | 51° N, 7° E | **3.5 ± 0.21** | 1988-2008 | - | yearly means |
| | | 4.7 | | - | yearly means (simplified approach) |
| | | 1 to 6 | | - | yearly means (for 11year subsets of time series) |
| Kalicinsky et al. (2016) | | **3 to 5** | 1988-2015 | - | yearly means |
| | | 5.0 ± 0.7 | | - | yearly means (multiple regression #1) |
| | | 4.1 ± 0.8 | | - | yearly means (multiple regression #2) |
| Kalicinsky et al. (2018) | | **3.5 ± 1.5** | 1988-2017 | - | summer means (May-Jul, m. r.) |
| | | 3.5 ± 0.8 | | - | winter means (Nov-Jan, m. r.) |
| Noll et al. (2017) | 25° S, 70° W | **4.7 ± 0.4** | 2000-2015 | - | UVES astron. telescope (individual spectra) |
| This study | 47° N, 11° E | 5.61 | 2009-2020 | 0 | yearly means, Feb – Jan (R²=0.61) |
| | | 6.20 | | 110 days | yearly means, Aug - Jul (R²=0.91) |
| | | **5.9 ± 0.6** | | **90 ± 65** | **concluding evaluation** |






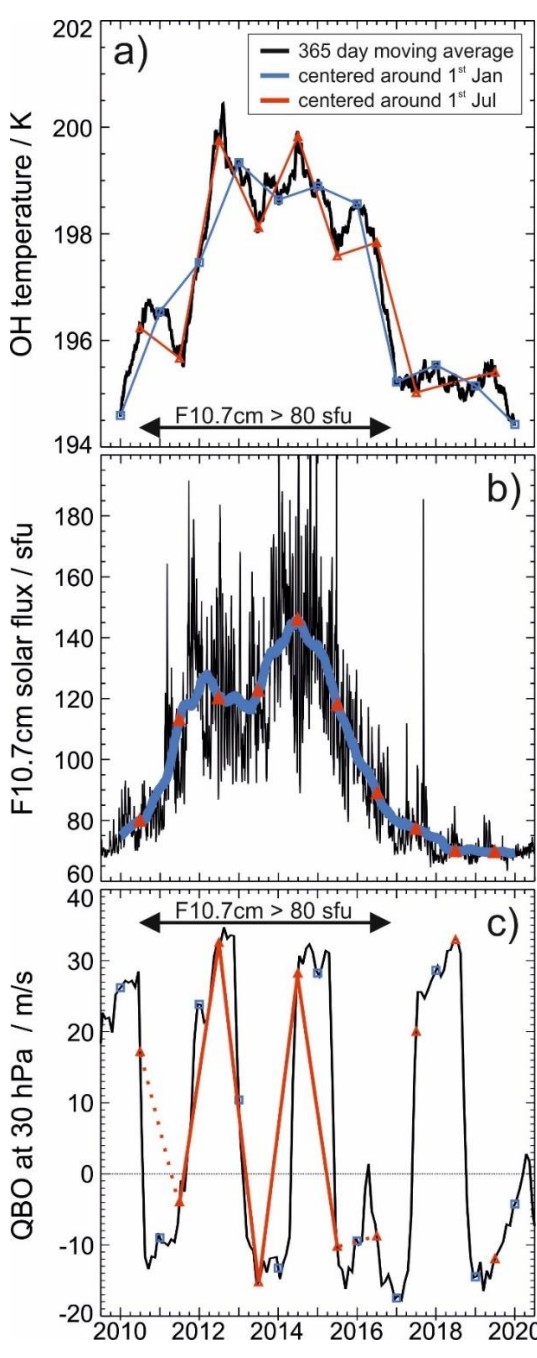

**Figure 11:** OH rotational temperature, F10.7cm solar radio flux and 30 hPa QBO winds (eastward: < 0m/s). Subset a) shows a 365-day moving average of the rotational temperature (black), the Jan-Dec yearly means (red triangles), identical to the solid blue curve in Fig. 9), and the Jul-Jun yearly means (blue). There is a strong biannual component best seen in the red and black curves. Subset b) shows daily values of the F10.7cm radio flux (black, source: NASA/GSFC's OMNI data set through OMNIWeb: https://omniweb.gsfc.nasa.gov), its 365-day moving average (blue) and Jan-Dec annual means (red triangles). Subset c) shows monthly QBO winds (black, source: https://www.geo.fu-berlin.de/en/met/ag/strat/produkte/qbo/index.html). Red triangles and blue squares mark the same dates as in subset a). Between 2011 and 2015 the phase of the QBO winds more less matches with the biannual cycle (highlighted by the solid red line). See text for further details.





A stable phase relationship between QBO and annual cycle might be required if the QBO is assumed to cause the observed modulation. Salby and Callaghan (2000) point out that the QBO tends to have shorter (longer) periods during solar maximum (minimum) conditions. This increases the probability for the phases of the two cycles to match for an extended period. Between 2011 and 2015 QBO phases at 30 hPa are indeed fairly stable. NH summers then subsequently fall into eastward and westward phases of the QBO. This relationship breaks in 2015/2016 and 2017/2018, when two subsequent NH summers first fall into

the eastward and then into the westward phase of the QBO (see red triangles in Fig. 11c)). At the same time the biannual signal in OH temperatures subsides. The QBO pattern observed in OH temperatures from 2011 until 2015/2016 also resembles the pattern of the semi-annual amplitude shown in Fig. 10b). Using satellite-based observations Xu et al. (2009) were able to show that the temperature diurnal tide exhibits strong modulations by both, QBO and the semi-annual oscillation, especially at heights above 80 km. In addition, the observed period of the temperature tide's amplitude modulation ranges between

approximately 22 and 25 months at these heights. Thus, the multi-year patterns observed in the OH temperatures might all be linked via systematic tidal modulations. Although we observe strong features of the semi-diurnal tide during the winter months (see Fig. 7a) for January) the discussion of the complex relation between F10.7cm, OH-T, QBO, semi-annual oscillation and atmospheric tides requires additional data and is beyond the scope of this study.

   As gaps have before been interpolated for the correct estimation of the annual means (s. Fig. 10), the long-term behavior of

the temperatures can now also be analyzed with spectral analysis methods requiring uninterrupted equidistant sampling. The wavelet analysis is such a method and it is rather practical for analyzing transient signals. Compared to the short time Fourier transformation (STFT) it can better produce a representation of the signal in frequency and time, because it fits short so-called mother wavelets (wavelike oscillations) to the data at different points in time. Here, we use the Morlet-Wavelet and the code by Roesch and Schmidbauer (2018) (see section "data and code availability"). Fig. 12 shows two wavelet power spectra of the

rotational temperatures for data versions 1.0 and 1.0A, with gaps filled by either the 2010-2019 mean for the respective missing date and alternatively with gaps interpolated by a combination of HA (annual and semi-annual amplitude) and MEM in its capacity as a linear prediction filter; shaded areas denote the cone of influence at both ends of the time series. Results for other combinations of data reduction scheme, gap filling and significance level are largely comparable and are shown in Fig. A3 in the appendices. Applying these different data reduction and analysis schemes helps in discriminating between features which

are due to artifacts.

   Several features are evident, besides the dominant annual cycle, e.g. the pattern of the semi-annual oscillation well reproduces the result shown in Fig. 10b) with the highest amplitudes in 2012 and 2014. A terannual component is present in some years (2013-2014 and 2016-2017) and more or less absent during other years, e.g. in 2018, which shows a strong 60-day oscillation instead (compare also Fig. 8b)). The quasi two-year oscillation can also be identified, but it shows up at periods below two

years at approximately 650 days or 21.5 months. Note that the time at which this oscillation is above the significance level is subject to the previous data reduction: Fig. 12a) shows it above the 95% significance level from late 2011 until early 2016 (matching the times discussed above). However, it can be considered significant from 2012 until the end of the time series in Fig. 12b) with longer periods before 2012. The period of ~21.5 months (~0.56 cycles per year / cpy) appears to be small. But





comparable periods of 0.59 cpy have before been reported by Salby and Callaghan (2000) and Salby (1997) in their
investigation of the QBO. They interpret the presence of this oscillation as the result of a non-linear interaction between the
29 months QBO (0.41 cpy) and the annual cycle (1.0 cpy).

This complex relation between OH rotational temperatures and the QBO regime complicates the study of the relationship
between solar flux and OH temperatures. In general, the solar flux appears to lead the OH temperatures by several months,
because it is increasing in 2011 and already decreasing at the end of 2015. The later increase of OH temperatures in 2012 can
595    be explained by the QBO eastward phase in 2011 causing lower OH temperatures. But this explanation does not apply to the
delayed temperature decrease in late 2016: solar flux decreases below 100 solar flux units (sfu) at the end of 2015 and the
QBO returns to eastward winds already in mid-2015, while OH-temperatures remain high at least until mid-2016.





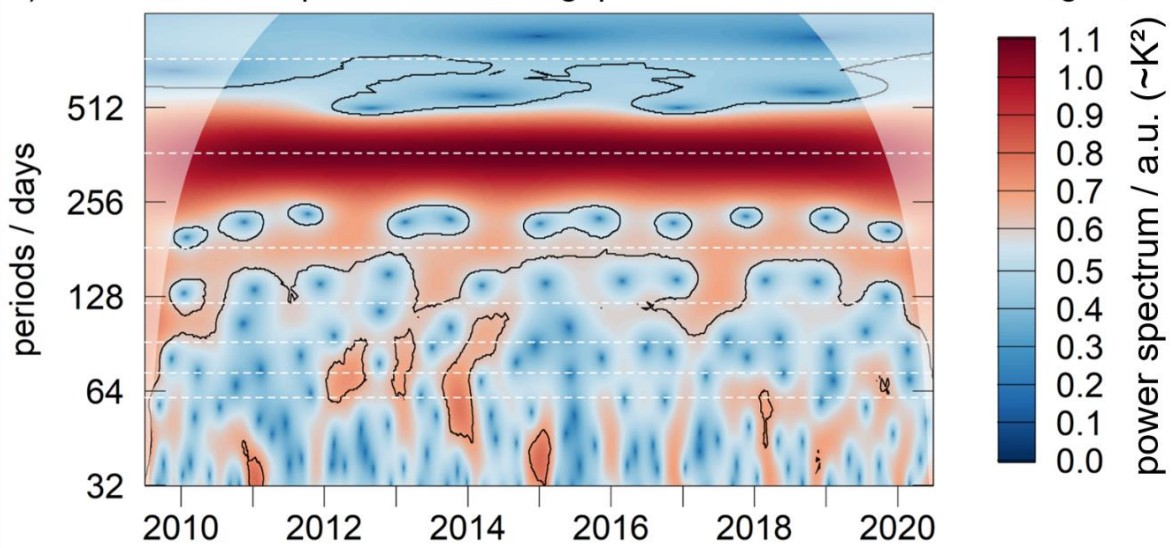

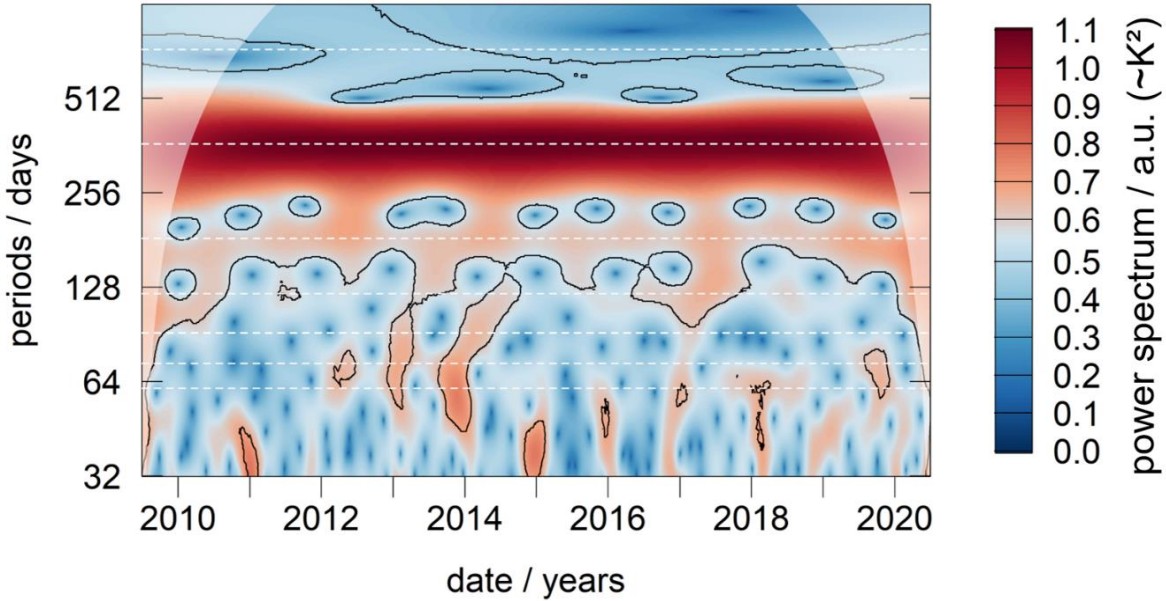

**Figure 12: Wavelet power spectrum of the UFS rotational temperatures (shaded areas indicate the cone of influence): a) data reduction version 1.0A, gaps have been filled with a combination HA (366 & 183 days) and MEM, black lines show 95% significance level; b) data reduction version 1.0, gaps have been filled with the 10-year mean for the missing date, black lines show 90% significance level; (alternative results concerning data version, gap filling and significance level are shown in Fig. A3 in the appendices). The dashed white lines highlight periods of 2, 1, 1/2, 1/3, 1/4, 1/5 and 1/6 years. Independent of the analysis, a quasi-biannual component is most significant between 2012 and 2016. However, it may also be present over the entire time at a period of less than two years (roughly 650 days / 21.5 months).**





Clearly, these relationships need further research, but this preliminary examination indicates that multiple regression analyses, which try to simultaneously minimize the influences of the solar cycle, QBO and other phenomena might lead to suboptimal results unless they account for a potential intermittency, (non-linear) interdependency and lag of some of the forcing mechanisms (see Table 3). Exact results often depend on details of the multiple regression models as was pointed out by Perminov et al. (2018), although Lednyts'kyy et al. (2016) showed how to minimize the degree of freedom by performing a step by step multiple-linear fit.

A potential lag between solar flux and OH temperatures has been discussed by French and Klekociuk (2011) and Holmen et al. (2014). They found that the correlation between these two parameters will improve, if lags of approximately 160 days or 65 days are introduced. Ammosov et al. (2014) briefly discuss again the QBO as a potential explanation for their (potential) lag of 25 months. Although rather apparent in their data they do not further emphasize this large lag lacking a proper mechanism explaining the large delay. All three data sets suffer from the fact that observations have to be suspended for a few months in summer due to the high-latitude observation sites; this clearly complicates certain analyses. However, as diurnal tides appear to be linked to the observed QBO and semi-annual oscillations (Xu et al. (2009)), either a time-delayed response of these tides or largescale mixing effects as discussed by Shepherd et al. (2000) might contribute to a time delayed downward transport and successive heating at OH heights.

The lag between OH-temperatures and solar flux is investigated in further detail in Fig. 13. Therefore, the black curve in Fig. 11a) is separated into 365 individual time series (i.e., one for each day of the year) composed of 10 yearly means (the red and blue curves in Fig. 11a) represent two examples). Thus, the individual data points of each 10-year time series are statistically independent from each other. The same is done for the solar flux. Then the Pearson correlation coefficient, R, is calculated for different time lags between the data sets. Fig. 13a) shows the value of $R^2$ as a function of the central date (day of year around which the OH-temperatures were averaged) and time lags of up to half a year (183 days), with the solar flux leading the temperatures. Due to the nature of the data the $R^2$ values are always high (see Figures 11 and 14 for comparison). Nevertheless, certain characteristics can be identified: 1) the correlation is higher between November and April ($R^2 > 0.85$), 2) the lowest correlation is obtained for August and small time lags ($R^2 < 0.70$) and 3) the highest correlation ($R^2 > 0.9$) is observed for time lags greater than sixty days in February or April. The highest (lowest) $R^2$-values of 0.91 (0.61) are obtained, if the OH-temperatures are averaged around 4th February (6th August) with a time lag of 110 (0) days. Fig. 13b) shows the respective estimates for the impact of the solar flux on the OH-temperatures. These vary between 5.4 K/100sfu and 6.5 K/100sfu. Apparently, higher correlation implies a higher influence of the solar flux. The highest values of 6.2-6.45 K/100sfu are obtained for large time lags (>70-170 days) in late July until the end of August, when the correlation reaches only average values between 0.75 and 0.85. It amounts to 6.20 K/100sfu on 4th February and 5.61 K/100sfu on 6th August. While calculating yearly means eliminates variations with periods up to and below the annual cycle, shifting these averages by half a year minimizes the influence of a potential biannual component. The phase relationship of such a biannual component with respect to the solar flux then determines the value of the observed lag.

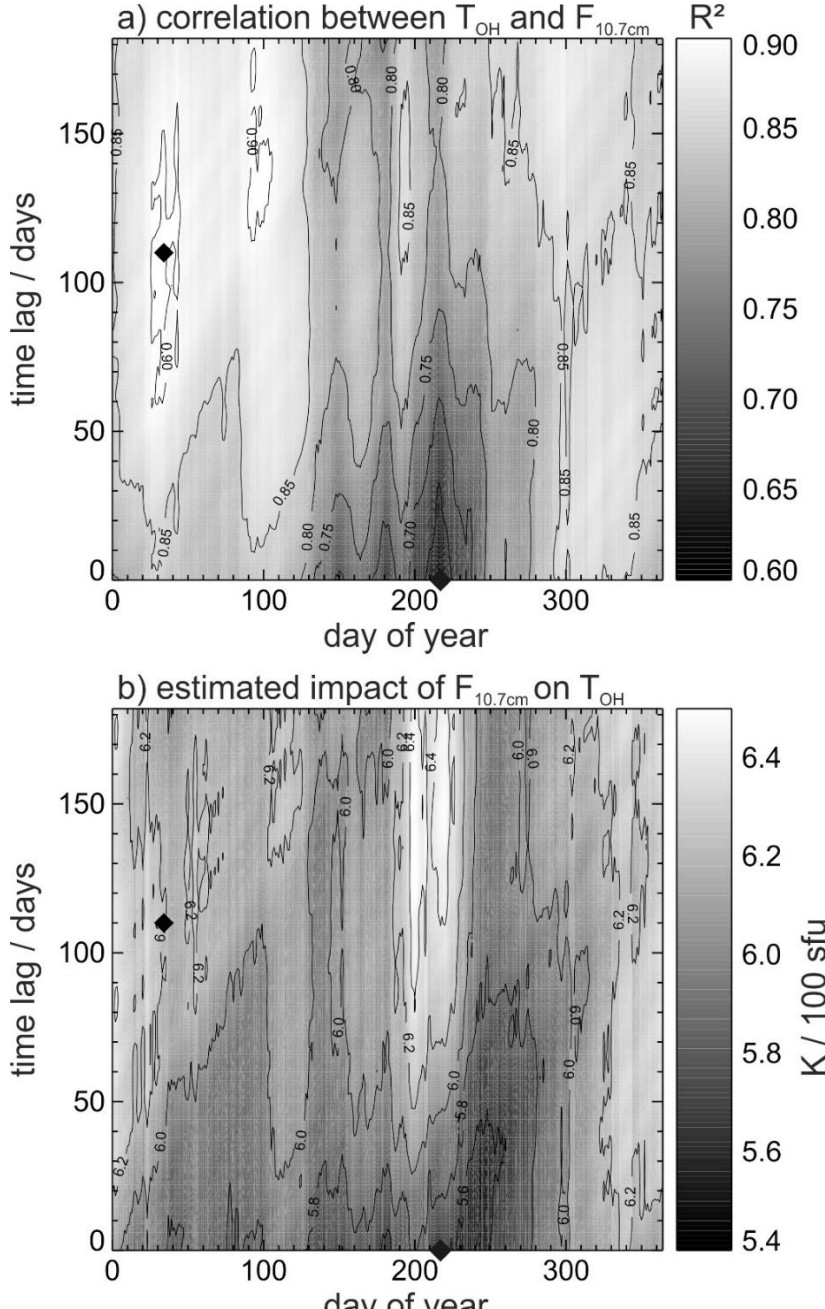


**Figure 13: Correlation between the yearly means of $T_{OH}$ and the solar radio flux. Subset a) shows the correlation (expressed by $R^2$) as a function of day of year (doy) and time lag; the highest correlation is achieved, when the yearly mean of $T_{OH}$ is centered around doy 1 to 120 (Jan to Apr) and F10.7cm is assumed to lead $T_{OH}$ by 80 to 160 days. Black diamonds denote days with the highest and the lowest correlation (4th February at 110-day time lag: $R^2=0.91$; 6th August, no lag: $R^2=0.61$). Subset b) shows the respective estimates for the impact of the solar flux on $T_{OH}$. It ranges between 5.4 and 6.45 K/100sfu. It amounts to 6.20 K/100sfu for the date of highest correlation and 5.61 K/100sfu for the date of the lowest correlation.**





For comparison, time lag and relation between $T_{OH}$ and F10.7cm are also obtained by calculating the cross correlation between the 365-day sliding averages of the nightly mean values shown in Fig. 11a) and b). The increased number of data points
(=3652) helps in judging the reliability of the previous results obtained from only ten well-defined yearly means, although this approach is mathematically not well-defined (since the individual data points are not independent). Here, the results are 5.92 (±0.05) K/100sfu up to 6.06 (±0.04) K/100sfu for time lags between 24 and 85 days with the correlation coefficients showing little variation from 0.780 to 0.782.

In summary, this shows a solar cycle influence of approximately 5.9 K/100sfu ± 0.6 K/100sfu at a time lag of 90d ±65d. The
lag is largely in agreement with the findings of French and Klekociuk (2011) and Holmen et al. (2014). A large uncertainty was adopted for the lag, as it does not appear to be constant over time (see Fig. 13a)). If the solar forcing is assumed to be indirectly taking effect on the OH temperatures via changes of middle atmospheric wind systems and associated waves, this appears to be more than reasonable since these are well known for their seasonality. This interpretation is further endorsed by the fact that correlation coefficients are clearly higher from November until April (winter/spring: doy 300 to 120 in Fig. 13a)).
It should be noted, that minimum $R^2$ values are found in August and the minimum forcing is found in September (dark areas in Fig. 13a) and b)). This is in remarkable agreement with French and Klekociuk (2011) although data and analysis methods differ substantially (see their Fig. 6). Clear evidences for different solar forcing magnitudes in summer and winter were also brought forward by Pertsev and Perminov (2008) and again confirmed by Dalin et al. (2020) with the response in winter being twice as large as in summer. But Kalicinsky et al. (2018) with their observing site being closer to UFS, do not see any difference
between winter and summer. A possible explanation lies in the time period covered by the time series. If the forcing is not constant over time, the results may vary, because authors are looking at different years. Offermann et al. (2010) argue that results depend on the length of the analysis window and windows should be substantially larger than the studied phenomenon. Holmen et al. (2014) on the other hand argue that although their analysis window meets this criterion, their result is close to 0K/100sfu for 1983-2001 and +10.9K/100sfu thereafter. Accordingly, they conclude with a reasonably large confidence
interval of 3.6 ±4.0 K/100sfu. Table 3 gives an overview of the magnitude of solar forcing on OH temperature time series obtained by various authors during the last decade (dec), including several alternative approaches concerning lags, data averaging, seasonal dependence and other parameters discussed by the authors.

Although solar cycle 24 is just barely captured by the measurements covered by the UFS data set and as several circumstances influence the results, there can be no doubt that temperatures roughly increased by 4 K (from 195 K to 199 K) in line with a
solar flux increase by approximately 70 sfu (see Fig. 11). Regarding the results obtained by other authors it is questionable if a respective increase of 8 K will be observed for a stronger solar flux increase of 140 sfu, which is frequently found for previous solar cycles. Only future studies can show, whether this is 1) due to other long-period oscillations superimposed onto the temperatures as proposed by Höppner and Bittner (2007) and elaborated on by Kalicinsky et al. (2018), 2) due to the solar forcing varying in time (e.g., Holmen et al. (2014)) or 3) due to non-linearity effects limiting the response in case of stronger
forcing.



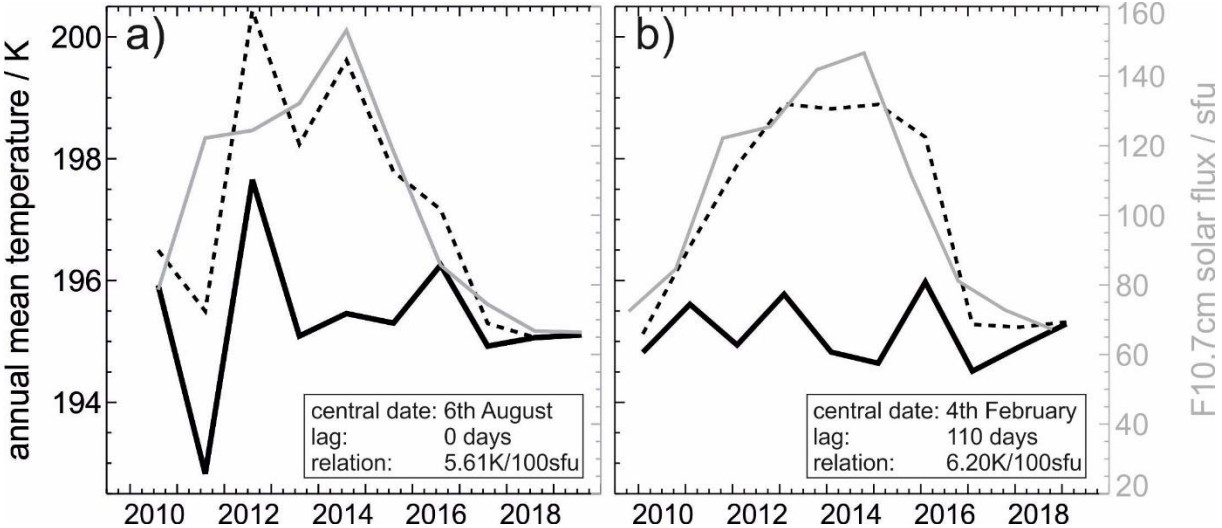

**Figure 14: Removing the influence of the solar radiation from the time series of OH temperatures. Subset a) shows the yearly means of $T_{OH}$ (dashed black) and F10.7cm (solid gray) centered around 6$^{th}$ August (no lag). The solid black line represents $T_{OH}$ referenced to 70sfu (assuming 5.61 K/100sfu), the 10-year change from 2010 to 2019 then amounts to -0.12K/dec ±1.42K/dec. Subset b) shows the same as a) but yearly means of $T_{OH}$ (dashed black) and F10.7cm (solid gray) were centered around 4$^{th}$ February and 17$^{th}$ October (110 day lag), respectively; now the 10-year change amounts to -0.14K/dec ±0.59K/dec (and 6.20 K/100sfu).**

Ultimately, an appropriate estimate for the solar forcing is required for a precise determination of long-term changes. Fig. 14 shows a comparison of two approaches based on the best and worst correlation obtained from Fig. 13a). Fig. 14a) shows yearly mean temperatures (dashed black) as a result of averaging from 7$^{th}$ February to 6$^{th}$ February (centered around 6$^{th}$ August) without any lag to F10.7cm (solid gray). Subset b) shows the same data, but temperature averages are calculated from 6$^{th}$ August to 5$^{th}$ August (centered around 4$^{th}$ February) with F10.7cm leading the temperatures by 110 days (averaged from 17$^{th}$ April to 16$^{th}$ April). The bold black lines result from applying the respective solar cycle forcing determined from Fig. 13b). Apparently, the resulting overall temperature change in the decade 2010-2019 is comparatively weak in both cases, but the latter approach significantly improves the confidence from -0.12K/dec ±1.42K/dec to -0.14K/dec ±0.59K/dec. This is mainly due to the fact that the temperature deviations in the years 2011 and 2012 are minimized by the latter approach. Since the averaging intervals are shifted by half a year, the latter approach indirectly accounts for the QBO forcing. But this QBO-signal is weaker during the last years with reduced solar flux (2017-2019). Therefore, any approach directly addressing the QBO-influence (e.g., as part of a multiple regression) should be done with great caution rather than assuming the QBO-influence on OH rotational temperatures was constant over time.





## 4 Summary and Conclusions

One decade of airglow observations at the NDMC-site UFS were examined with respect to data quality as well as certain aspects of the retrieval and their influence on later results. Advantage was taken of the fact that UFS is equipped with two instruments, which were operated in an identical setup for most of the time and with slightly different setups during some

years.

The main findings concerning technical aspects, such as data quality and the impact of certain aspects of the retrieval, on the results are:

1)   The two-instrument setup reliably prevents the occurrence of larger gaps in the data set due to technical failures of one instrument and nightly mean values can be obtained for 3382 of 4018 nights. However, the remaining gaps are

710       not randomly distributed in the data set. At UFS there are systematically fewer observations in May and June due to an increased number of convective clouds. At the same time MLT temperatures reach their annual minimum values. Consequently, estimates of annual or summer mean temperatures must account for this fact. The maximum respective sampling bias for the annual mean is estimated to be +0.8 K in 2015 (see Fig. 9a).

2)   Two methods have been applied in filling the remaining gaps: a) the 10-year average temperature for each missing date and b) a combination of HA (with 365 and 182.5-day period) and MEM to account for the short-term variability. These perform equally well in correcting the annual mean (compare Figures 9c and 9d). In addition, these methods perform also well in correcting the time series of the individual instruments, which exhibit larger gaps due to technical failures (see Figures 9b and 9c).


3)   An improved method for deriving the nightly mean (and its uncertainty) from a given set of observations was developed in order to better estimate precision, accuracy and the probable sampling bias for each night individually. This method is mainly meant to better estimate the influence of sun-synchronous atmospheric tides and avoids empirical constraints (such as a minimum observation time required for a nightly mean to be considered

725       representative). This nightly sampling bias is found to be up to 4 K. However, nights with positive and negative sampling biases are equally distributed throughout the year and there is little influence on the annual mean (±0.2 K). Shorter averages like monthly means might however be influenced by this sampling bias (see Figures 7, 8).

4)   The choice of Einstein-A-coefficients is well-known to influence the temperature retrieval. The analysis of all sets of

730       coefficients still in use, from Mies (1974) to Brooke et al. (2016), reveals that the temperature differences depend on the absolute temperature (see Fig. 3). This in turn means that not only estimates of annual means will differ between different retrievals but also estimates of the annual cycle's amplitudes will differ significantly. In case of the two mentioned sets the annual amplitudes differ by 0.4 K (see Fig. 10).





The improvement of data quality by the parallel operation of two instruments is only a means to an end: increasing the confidence of scientific analyses concerning atmospheric dynamics and especially long-term changes of OH temperatures. The main findings in this context are:

1) The amplitude of the annual oscillation is four to six times larger than the semi-annual amplitude (17 K to 18.5 K
compared 2.5 K to 4 K). If the FoV of both instruments are only slightly separated (~100 km), both oscillations will become apparent in the differences of the respective time series (see Fig. 5). This implies that both oscillations experience a strong gradient at UFS (47.42° N, 10.98° E). This needs to be considered when comparing data between sites, which are separated by larger distances, or when data are compared to satellite observations with respective miss distances.


2) Quantitative estimates of the solar cycle response critically depend on the chosen averaging intervals and on the time lags between solar flux and OH temperatures. The highest correlation ($R^2$=0.91) is achieved, if annual means of OH temperatures are centered around 4th February and solar flux is assumed to lead OH temperatures by 110 days resulting in a response of 6.20 K/100 sfu. The lowest correlation ($R^2$=0.61, 5.61 K/100 sfu) is achieved, if annual
means of OH temperatures are centered around 6th August without any time lag. The cross-correlation analysis of 365 d running means leads to a comparable result of 5.9 to 6.1 K/100sfu for time lags between 24 and 85 days ($R^2$=0.78). In summary, a solar forcing term of 5.9 ±0.6 K/100sfu covers the observations during solar cycle 24 well.

However, with respect to long-term changes assuming a response of 6.20 K/100 sfu and 110 days time lag
significantly increases the confidence of the 10-year linear change (2010-2019) from -0.12K/dec ±1.42K/dec to -0.14 K/dec ±0.59K/dec (see Fig. 14).

3) Between 2011 and 2016 temperatures show a distinctive quasi-two-year periodicity. During QBO westward phases in 2011, 2013, 2015 the temperature decreases by approximately -1 K and during QBO eastward phases it increases
by +1 K in 2012 and 2014. This appears to be linked to a time, when the QBO showed periods closer to 24 months (see Fig. 11). This clear relationship disappears after 2015, when the period of the QBO increases again and solar flux is reduced. But wavelet spectral analyses show a quasi-biannual signal being constantly present, albeit at a lower period of only approximately 21.5 months (~0.56 cpy) becoming more pronounced between 2011 and 2016.

This hints at a complex interplay between solar cycle, QBO and mid-latitudinal dynamics at MLT heights, with the observed modulation of ~0.56 cpy probably being the result of a coupling between the annual cycle (1 cpy) and the climatological average QBO period (~0.43 cpy) as proposed by Salby and Callaghan (2000) and Salby (1997). On



the other hand, at MLT heights the QBO might only act as a kind of switching mechanism either amplifying or reducing the impact of other forcings which are strictly bound to the annual seasonal cycle.


4) The semi-annual cycle shows a similar behavior with the highest amplitudes of 4 K observed in 2012 and 2014 and the lowest amplitudes around 2.5 K observed in 2011, 2013 and 2015. Unlike the biannual signal, the correlation between QBO-phase and amplitude of the semi-annual oscillation appears to continue after 2015, with (slightly reduced) amplitudes of 3 K in 2016 and 2018 and (slightly increased) amplitudes in 2017 and 2019.


5) The year 2016 stands out in several aspects: it marks the transition from high to reduced solar activity and for the first time since 1953 the QBO experienced a disruption of its usual behavior in 2015/2016 (e.g., Newman et al. (2016)). It is also remarkable in the UFS OH temperature time series, best seen in the phase of the annual cycle (Fig. 10c), but it also marks the end of the previously pronounced two-year periodicity mentioned above (black and red curve in Fig.

11a).

The main focus of this study is on data quality aspects, taking advantage of the fact that UFS is equipped with two identical instruments avoiding large data gaps. The subsequent analysis of the dataset itself deliberately relied on straightforward methods revealing strong indications that a QBO-related signal is maximized during enhanced solar activity but less

pronounced during the current solar minimum (since 2016). This interdependency of MLT dynamics with QBO, solar forcing, seasonal cycle, long-term trend and other phenomena (e.g., La Niña, El Niño, …) indeed needs more research.

So far, the conservative analyses already performed indicate that at least some forcing mechanisms may have a time-dependent component or exhibit some intermittency not well represented in common multiple regression approaches. Next steps will therefore include the integration of other observing sites equipped with similar instruments and the application of more

sophisticated analysis methods including the variability of airglow intensities.





# Appendices

Table A1: The upper part shows the number of nights with successful data acquisition (including nights with cloud coverage); comments refer to the number marked by an "*". The lower part lists the remaining large gaps of the final data product, which
extend for at least seven successive days, and the respective cause.

| year | number of nights | | Comment |
| --- | --- | --- | --- |
| | GRIPS 5 / GRIPS 8 | GRIPS 7 | |
| 2008 | 56 | - | 2008-10-25: 1st light GRIPS 5 |
| 2009 | 209 | - | many computer related issues (RAM, USB, …) |
| 2010 | 236 / 119 | 340 | 2010-01-20: start observations with GRIPS 7<br>2010-10-01: GRIPS 8 replacing GRIPS 5 |
| 2011 | 360 | 363 | - |
| 2012 | 366 | 366 | - |
| 2013 | 360 | 351* | 2013-09-11, 9 nights: shutter closed |
| 2014 | 333* | 363 | 2014-08-23, 25 nights: filterwheel failure |
| 2015 | 341* | 357 | 2015, 24 nights (intermittent gaps): computer related issues |
| 2016 | 363 | 319* | 2016-06-23, 26 nights: computer failure<br>2016-12-10, 21 nights: filterwheel failure |
| 2017 | 359 | 323* | 2017-01-01, 29 nights: filterwheel failure contd., + intermittent computer failures |
| 2018 | 362 | 359 | - |
| 2019 | 297* | 353 | 2019-08-11, 68 nights: filterwheel failure |
| 2020 | 364 | 364 | - |

| remaining large gaps in final data product | | |
| --- | --- | --- |
| year | cause | number of missing nights |
| 2009 | cloud coverage | 2009-10-08: 10 nights |
| 2009 | computer related | 2009-12-20: 14 nights |
| 2013 | UFS power failure | 2013-08-13: 7 nights (power failure: 5 nights) |
| 2019 | cloud coverage | 2019-01-02: 5 & 7 nights (12 of 13 nights missing with one observation night in-between) |






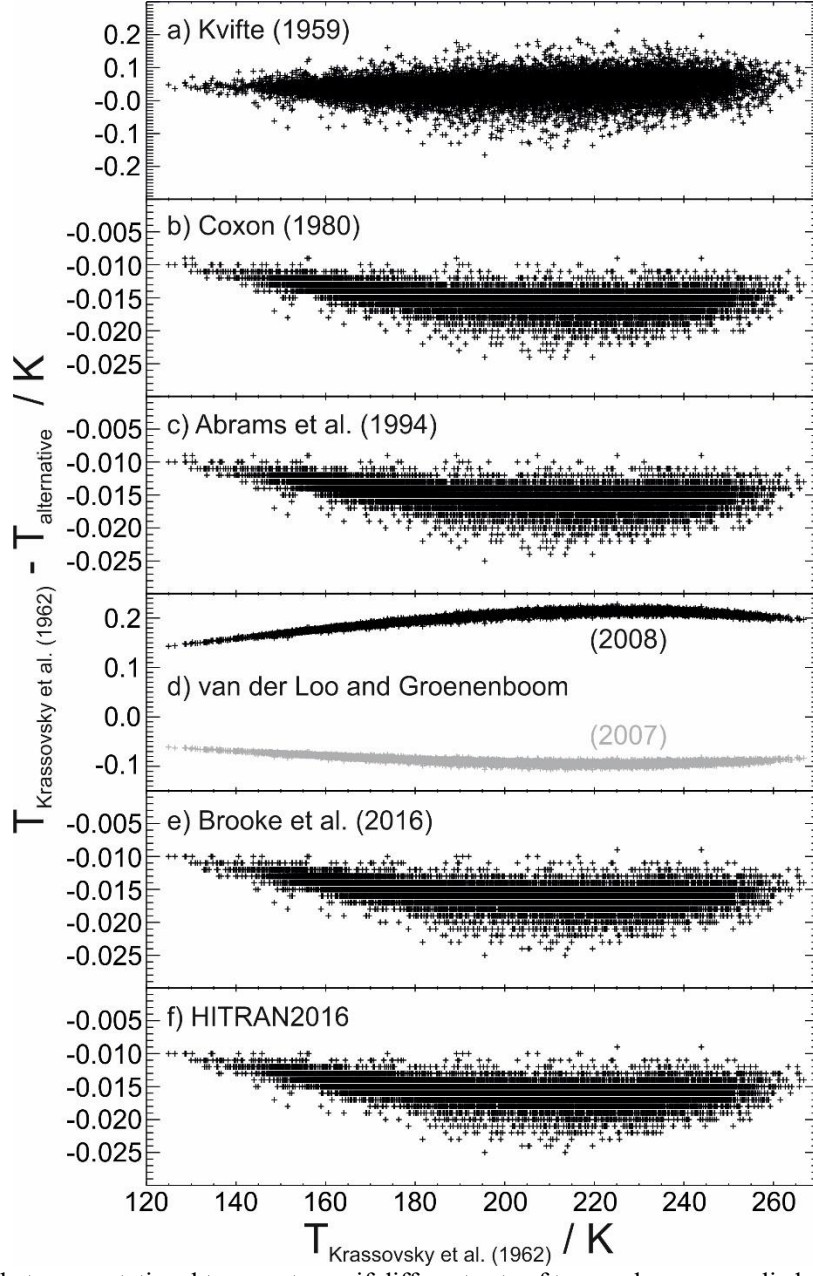

Figure A1: Differences between rotational temperatures, if different sets of term values are applied during the (otherwise identical) retrieval. The temperatures are obtained from the same spectra as those shown in Fig. 3. The influence of the term values is one order of magnitude smaller than the influence of the Einstein A coefficients. Note: horizontal white lines in some panels are caused by the limited resolution of 0.001 K.






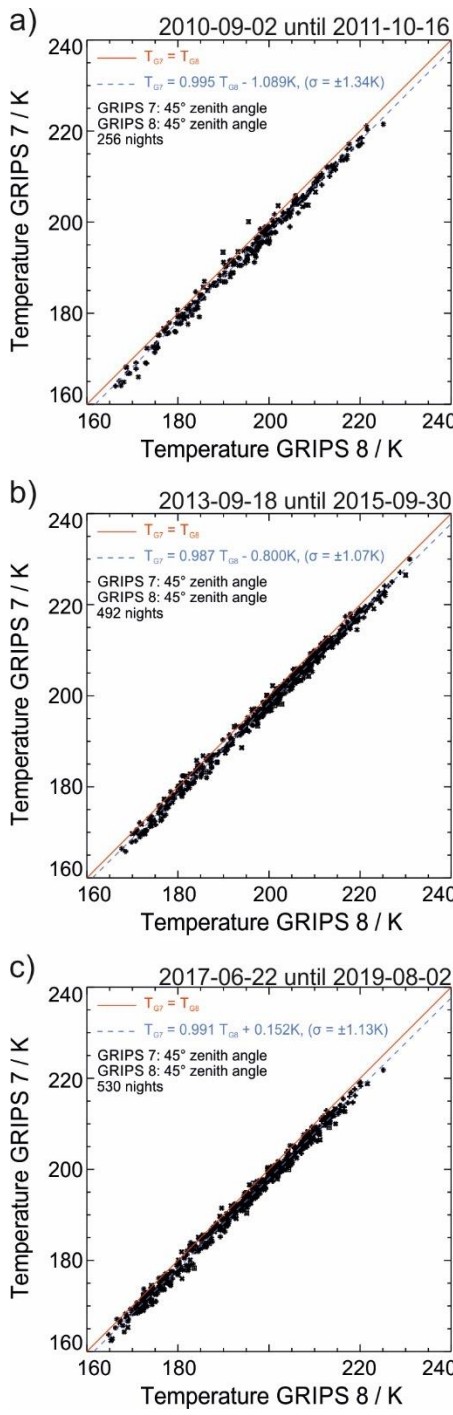

Figure A2: Scatterplots of GRIPS 7 vs. GRIPS 8 temperatures for the individual time periods with matching fields of view of both instruments (bisecting lines: solid red, fit to the data: dashed blue). The mean temperature dependence of $T_{G7} = 0.991$ K*$T_{G8}$ (see Fig. 4) is well reproduced with individual values of 0.995, 0.987 and 0.991.

**Figure A3: Results of the wavelet spectral analysis for eight possible combinations of time series data version (left column: v1.0A, right column: v1.0), method of gap filling (spectral methods HA & MEM in panels a), b), e,) f) or insertion of respective 10y-mean value in panels c), d), g) and h)) and significance level of the spectral power (upper four panels: 95%, lower four panels: 90%, indicated by the black lines). Shaded areas indicate the cone of influence; panels a) and h) are identical to Fig. 12. Most features appear in any of these analyses. A precise identification of the oscillation with a period close to two years is difficult but clearly below 24 months (top dashed horizontal white line).**



**Code and Data availability**

The OH temperature time series produced and analyzed in this study are available according to the standard conditions outlined by NDMC. Both processing versions v1.0 and v1.0A have been archived at the World Data Center for Remote Sensing of the Atmosphere (WDC-RSAT). They are traceable under the following digital object identifiers (DOI): 10.26042/WDCRSAT.XZB5TZQG (UFS_v10) and 10.26042/WDCRSAT.Y0AOE0PZ (UFS_v10A).

The wavelet analyses were performed with the R-package 'WaveletComp': Angi Roesch and Harald Schmidbauer (2018).
WaveletComp: Computational Wavelet Analysis. R package version 1.1. https://CRAN.R-project.org/package=WaveletComp.

**Author contribution**

Both, manuscript and formal analyses contain valuable input from all authors. Carsten Schmidt performed most of the analysis, he prepared the original manuscript including visualization and took care of validation and data curation for more than a
decade. Lisa Küchelbacher contributed parts of the manuscript and performed most of the spectral analyses including visualization of the results. Sabine Wüst contributed parts of the analysis; especially, she investigated the consequences of changing the FoVs of the instruments. She was also deeply involved in project administration and funding acquisition. Michael Bittner initiated the observations at UFS including the concept of parallel observations with two instruments. Also, he stimulated the investigation of certain parameters taking influence during the processing (e.g., Einstein-A-coefficients). In this
role he was responsible for conceptualization, funding acquisition, project administration and supervision of the observations.

**Competing interests**

The authors declare that they have no conflict of interest.

**Acknowledgements**

The airglow observations at UFS "Schneefernerhaus" are funded by the Bavarian State Ministry for Environment and
Consumer Protection (BayStMUV projects GUDRUN, grant no. 71b-U8729-2003/125-13; GRIPS3 Back-Up, 2009/40051; BHEA, TLK01U-49580; LUDWIG, TUS01UFS-67093; VoCaS, TKP01KPB-70581; AlpEn-DAC: TUS01UFS-72184). The authors wish to thank Dr. Stefan Noll and Dr. Oleg Goussev for their valuable input concerning OH Einstein coefficients and for Dr. Goussev's assistance with long-term archiving and referencing the data at the World Data Center for Remote Sensing of the Atmosphere (WDC-RSAT).

We acknowledge the use of NASA/GSFC's Space Physics Data Facility's OMNI data set through OMNIWeb: https://omniweb.gsfc.nasa.gov, where we obtained the F10.7cm values used in this study, accessed on 8 September 2020. The QBO data have been obtained from Freie Universität (FU) Berlin: https://www.geo.fu-berlin.de/en/met/ag/strat/produkte/qbo/index.html, accessed on 6 July 2021.





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
