# Peer review of "OH airglow observations with two identical spectrometers: benefits of increased data homogeneity in the identification of the 11-year solar cycle-, QBO-induced and other variations"

_Atmospheric Measurement Techniques, 2023_

## Referee Comment (RC1)

**Manuscript number: AMT-2023-61**
OH airglow observations with two identical spectrometers: benefits of increased data homogeneity in the identification of the 11-year solar cycle-, QBO-induced and other variations

by Carsten Schmidt, Lisa Küchelbacher, Sabine Wüst, and Michael Bittner

**Overall Summary:**

The manuscript present the result obtained by a decade observation of OH(3-1) rotational temperature from UFS(47.42° N, 10.98° E) using two identical Spectrographs to avoid data gaps. In addition to the instrument failures the nightglow observation suffers from data gaps in summer month due to convective clouds. The authors deal with such data gaps and also issues related to the different observational duration, number of data sampling per nights, and time interval over which the observations are available to get the reliable nightly mean values. They have come up with the improved calculations of nightly and annual means using a decade average temperature for missing date as well as HA and MEM which take cares of short-term variations (GWs, Tides). They have then investigated AO, SAO, QBO, and solar cycle influences using nightly and annual OH(3-1) rotational temperature mean value.

The quality of the data, observation duration, and reliable calculation of the nightly mean is impressive and such data sets are rare in the world. Such data sets are required to understand seasonal influences, solar forcing, MLT dynamics, and long term trend in MLT region. After minor revision as suggested below this manuscript is strongly recommended for the publication in AMT.

**Specific comments:**

1. Page 6 line 128-135: Whether authors are talking about intensity calibration? What decides long term stability of the spectrographs? How intensity calibration affects the accuracy of derived rotational temperatures? Since InGaAs are known for their large dark currents getting a perfect dark frame is also a challenge which affect accuracy of the temperature measurement. Whether dark frames are generated daily?

2. Section 2.3: As it can be seen from equation 2.1 the derived temperature mainly depends on Einstein-A-coefficients and earlier literatures showed that it is less dependent on the term values. It is not clear to me the purpose of Figure 3 as this this expected, unless we use different set of A values and compare with an independent measurement and show which A values provide rotational temperatures that are close to that independent measurement which can be considered to be ambient temperature of the OH emission altitude (~87 km). In further analysis considering different A's will just add a constant value of the derived parameters (e.g. temperature values, amplitudes etc.).

3. Figures 4a and 4b: why the linear relationship matches well with MSISE in Figure 4a and does not match in Figure 4b?

4. Figure 5: Whether GRIPS 8 and GRIPS 7 data separately will show AO and SAO? Since, if AO and SAO are present in the rotational temperature data, it should be visible in

individual measurement as well so in OH intensities (e.g., Singh and Pallamraju, Ann. Geophys., 2017, doi:10.5194/angeo-35-227-2017).

**Minor comments:**

1. Page1 line 22: How the precise value depends on details of the analysis?
2. Page 2 line 55-56: sentence may be modified like→ If the instrument cannot be repaired it can be replaced with a similar new instrument.
3. Page 5 line 104 to resolve the P1-lines→ to resolve the first three P1-lines (as the first three P1 branch lines are used for the rotational temperature determination, so contamination from nearby P2 branch should not be there).
4. Page 6 line 116: I am just curious to know why GRIPS instruments have been operated with an oblique FoV? Since different FoV will see different part of the sky which needs to be taken care in the data analysis and interpretation of the results. What are the challenges in operating GRIPS in vertical direction in future?
5. Page 6 line 125: How a minor misalignment affects data cadence?
6. Page 9 line 223: observed volumina → observed volumes
7. Page 9 line 224; Wüst et al (2016) → Is it Wüst et al. (2017)?
8. Page 11 line 243: while keeping the 180° (tward) azimuth angle. Subset a) → while keeping the azimuth angle of 180°.
9. Figure 4: It is difficult to see dashed green triangle.
10. Page 11 line 43: Subset a) of Fig. 4 → Fig. 4a) here and elsewhere. e.g., Page 22 L 247: Subset b) → Fig. 4b)
11. Figure 5 caption: annual and semi-annual oscillation → annual and semi-annual oscillations; Harmonic Analysis → harmonic analysis
12. Page 15 line 348: Sorry for my ignorance but how authors have arrived at the true value of ΔT is often closer to ±2 K (or ±2.75 K/√2)?
13. Section 3.4 and Figure11: Replace quasi-biannual with quasi-biennial wherever it appeared in the text.

---

## Author Comment (AC1)

**Reviewer #1:** https://doi.org/10.5194/amt-2023-61-RC1

**Manuscript number: AMT-2023-61**

OH airglow observations with two identical spectrometers: benefits of increased data homogeneity in the identification of the 11-year solar cycle-, QBO-induced and other variations

by Carsten Schmidt, Lisa Küchelbacher, Sabine Wüst, and Michael Bittner

**Overall Summary:**

The manuscript present the result obtained by a decade observation of OH(3-1) rotational temperature from UFS($47.42°$ N, $10.98°$ E) using two identical Spectrographs to avoid data gaps. In addition to the instrument failures the nightglow observation suffers from data gaps in summer month due to convective clouds. The authors deal with such data gaps and also issues related to the different observational duration, number of data sampling per nights, and time interval over which the observations are available to get the reliable nightly mean values. They have come up with the improved calculations of nightly and annual means using a decade average temperature for missing date as well as HA and MEM which take cares of short-term variations (GWs, Tides). They have then investigated AO, SAO, QBO, and solar cycle influences using nightly and annual OH(3-1) rotational temperature mean value.

The quality of the data, observation duration, and reliable calculation of the nightly mean is impressive and such data sets are rare in the world. Such data sets are required to understand seasonal influences, solar forcing, MLT dynamics, and long term trend in MLT region. After minor revision as suggested below this manuscript is strongly recommended for the publication in AMT.

Thank you very much for this benevolent assessment of our study. We found your comments very valuable and have corrected all errors. Also, we rephrased certain instances that may have been hard to understand. We hope that we succeeded in answering all your questions. In those few cases, in which we did not apply changes to the manuscript, we took special care to provide detailed answers to your comments, hoping that the answers clarify why a detailed discussion of the respective topic is beyond the scope of the current manuscript.

Below, we provide the answers to comments in blue and respective changes in the manuscript in green. Comments of the reviewer remain black, as do those parts of the manuscript, that stay unchanged. Pages and line numbers refer to the annotated version of the manuscript.

**Specific comments:**

1. Page 6 line 128-135: Whether authors are talking about intensity calibration? What decides long term stability of the spectrographs? How intensity calibration affects the accuracy of derived rotational temperatures? Since InGaAs are known for their large dark currents getting a perfect dark frame is also a challenge which affect accuracy of the temperature measurement. Whether dark frames are generated daily?

These are several good questions indeed, which we address one after the other. Note: the process of calibration and dark current subtraction has been discussed in substantial detail by Schmidt et al. (2013), DOI 10.1016/j.jastp.2013.05.001, who describe the identical instrument GRIPS 6 and the identical data version 1.0. We do not aim at reproducing this discussion in the current manuscript, as it would not add anything new here. As can be taken from the following pages, these reasonable questions require a detailed discussion to be answered correctly.

1.1 Whether authors are talking about intensity calibration?      /
1.3 How intensity calibration affects the accuracy of derived rotational temperatures?
*(see next paragraph for "1.2 What decides long term stability of the spectrographs?")*

The rotational temperatures are retrieved from intensity ratios of several $P_1$-lines: In order to retrieve correct temperatures, the instruments must of course be calibrated ensuring that the observed lines can be compared (i.e. it must be excluded that the instrument receives 90% of the photons from one line and only 80% from another line). Thus, the relative spectral characteristics of the spectrometer response are crucial for the temperature retrieval.
The important lines are separated by only ~19 nm (1524 nm to 1543 nm) and a zero-order approximation may be that the spectral characteristics of the instruments do not change significantly over this distance. Anyway, we compared the spectrometers' response to calibrated lamps, displayed in the following Figure R1.1.
The calibration with lamps is rather complex, as these are orders of magnitude brighter than the airglow (0.001s - 0.01s exposure times during calibration versus 15s at nighttime airglow observations). They require the long-pass filter in the setup to suppress higher grating orders (the maximum of the lamp output at 800nm coincides with the $2^{nd}$ order maximum at the same spatial direction as the $1^{st}$ maximum of 1,6μm). And they require a diffuser to fill the entire field of view etc.pp.. Thus, we only derive the relative responsivity change and do not attribute absolute units to the derived intensities yet.
Concerning the accuracy, one can determine the relative responsivity at the three critical lines (1524nm, 1533nm and 1543nm) from Figure R1.1. They amount to approximately 1.010, 1.010 and 1.007. Thus, $P_1(4)$ is systematically underestimated by GRIPS 8 compared to the other two lines. If this not accounted for in the retrieval, temperatures are systematically underestimated. In this case, instead of 200 K, we would retrieve 199.6 K with our algorithm (see also answer to question 1.2).

[Figure]

*Figure R1.1: upper panel: signal strength recorded by GRIPS 8 (blue) versus nominal calibration source output (yellow) as a function of wavelength. Lower panel: relative spectral responsivity (blue) and polynomial fit (black). The responsivity in the wavelength range critical for the temperature retrieval (between 1524nm to 1543nm) is relatively flat.*

1.2 What decides long term stability of the spectrographs?
We consider the following four elements of the systems to be the most critical for the long-term stability:

1. The InGaAs-detector itself,
2. The coating of mirrors and grating,
3. The long pass filter at the entrance and
4. The laboratory window

At the start of the observations in 2008 we had little experience with InGaAs, since GRIPS 1 to GRIPS 4 were based on Ge photodiodes. These older detectors didn't change their spectral characteristics in 25 years of operation. Today, we know that the InGaAs grating spectrometers are also rather stable. During the development of improved calibration approaches with the colleagues of the German Metrology Institute, PTB, we learned that their institute operates several grating spectrometers for the calibration of lamps. This backs up our observation during our early re-calibration with lamps (every 6 months), that the spectrometers appeared to more stable than the lamps and much easier to handle. Thus, between 2015 and 2017 we switched from tungsten lamps to large area thermal radiators for calibration (operated at 100°C-120°C).

Coatings of the mirrors and grating are standard aluminum and usually quite stable. However, the spectral properties of the long pass filter at the entrance may change over time. In order to slow its aging a shutter protects it from the exposure to direct sunlight during daytime. The laboratory windows on the other hand are exposed to harsh environmental conditions (thermal stress and UV radiation). They are made from float glass with reduced impurities. Figure R1.2 shows the respective transmission curve.

Up to this day we were unable to detect any significant drift in the optical properties of any of the optical components or of the retrieved temperatures,

[Figure]

*Figure R1.2: Transmission of the laboratory window. The temperature retrieval depends on the line ratio of emission lines at different wavelengths. Any optical element that exhibits a slope in spectral transmissivity (or reflectivity) can influence the accuracy of the retrieval. The lab window is often overlooked in this context, because it is not always easy to calibrate. Also, it is exposed to environmental conditions with the potential to alter its properties. No such changes have been observed so far at UFS.*

1.4 Since InGaAs are known for their large dark currents getting a perfect dark frame is also a challenge which affect accuracy of the temperature measurement. Whether dark frames are generated daily?

The high dark currents of InGaAs are indeed the main challenge in operating the spectrographs. We spent a significant amount of time in developing an approach to correctly assess these dark currents. This approach was described in detail by Schmidt et al. (2013), DOI 10.1016/j.jastp.2013.05.001.

Challenges and solutions can be summarized as follows: typically, dark currents (DC) are higher than the OH signal by a factor of fifty. Correctly removing these is possible, because they are rather stable and only exhibit a small drift, which correlates with external (i.e. lab and spectrograph) temperature. However, as their noise approximately scales with sqrt(DC), the dark current is the primary source of noise. Thus, dark currents are acquired twice every fifteen minutes resulting in ~50 to ~100 dark frames per night. These are averaged, rescaled and interpolated to derive an individual value for each pixel for each point in time. The left panel in Figure R1.3 shows the raw data (black) of one spectrum together with the result of the respective dark current assessment (red). The right panel shows the same spectrum after subtraction of dark currents and application of all calibrations. Despite the large DC only a small background (blue) remains. This remaining background originates at the sky, because it vanishes during phases of dense cloud coverage and is not correlated with the bright airglow lines. It probably contains varying contributions from airglow continuum emissions, scattered moonlight as well as some scattered radiation during twilight.

[Figure]

*Figure R1.3: left panel: raw data (black) with dark current (red); right panel: the same spectrum after application of all calibrations. To a large degree the remaining background (blue) originates at the sky (concluded from: 1) analysis of cloud-covered spectra, when the residual background drops to zero and 2) investigation showing this residual background is not correlated with the bright airglow lines).*

2. Section 2.3: As it can be seen from equation 2.1 the derived temperature mainly depends on Einstein-A-coefficients and earlier literatures showed that it is less dependent on the term values. It is not clear to me the purpose of Figure 3 as this this expected, unless we use different set of A values and compare with an independent measurement and show which A values provide rotational temperatures that are close to that independent measurement which can be considered to be ambient temperature of the OH emission altitude (~87 km). In further analysis considering different A's will just add a constant value of the derived parameters (e.g. temperature values, amplitudes etc.)

We agree. As is correctly pointed out by the reviewer, the application of different A coefficients to OH airglow observations has been studied before and it is important primarily when comparing one observation to the other. It is true, that the influence of different sets of Einstein-A-coefficients is known in the airglow community and that their influence is stronger in the OH(6-2)-retrieval than in in the OH(3-1)-retrieval (e.g., French et al. (2000), DOI: 10.1007/s00585-000-1293-2).

It is however our impression that the full consequences these coefficients have on the retrieval, is not well established. This concerns especially the temperature dependence displayed in Figure 3. Moreover, the successive influence of this temperature dependence on the amplitude of the annual oscillation is rarely documented (see Figure 10). Therefore, we find it to be an interesting result of our investigation that adding a constant to the annual amplitude is actually sufficient when comparing analyses relying on different Einstein coefficients. Due to the variability of the annual cycle, it is important to note that a constant will do this job (which is due to the combination of the small temperature dependence displayed in Figure 3 and the small variability of the annual cycle). Furthermore, the semiannual oscillation is not affected at all (at our site), which is reasonable but it wasn't guaranteed a priori that this is really the case.

Compared to the older publications on this topic, we also thought it would be worthwhile to include the more recent coefficients by Brooke et al. (2016), which have been used by some researchers in recent years and to provide an overview of how they influence the retrieval compared to the more established ones.

Moreover, fifteen years ago, the airglow observations at UFS were started as part of an initiative to establish UFS as a so-called ground-truthing center for a number of satellite-based observations (concerning mostly tropospheric/stratospheric trace gases, which are difficult to retrieve from nadir observations over mountainous terrain, but also MLT parameters such as airglow). Thus, the analysis of the A-coefficients is part of our core aim to provide a data set, to which anyone can compare their own data with reasonable high accuracy. This includes documenting the main influences on the retrieval and certainly the way Einstein-A-coefficients influence this retrieval and the precise magnitude airglow temperatures and important successive parameters (and why) are part of this endeavor.

3. Figures 4a and 4b: why the linear relationship matches well with MSISE in Figure 4a and does not match in Figure 4b?

Two factors influenced the investigation of the second time period, shown in Figure 4b:

1. The overall difference predicted by MSISE is of the order of ~0.5 K (min to max). It's significantly smaller than for the time period displayed in Figure 4a (~1.5 K min/max). This is reflected by the slope of 1.01, which differs less from the ideal 1.0 than the 0.96-slope in Figure 4a. This is small compared to the scatter of the observational data (see also Figure 5) and hard to identify even with good observational coverage, but:

2. The quantitative comparison of the data in Figure 4b is further complicated by two large data gaps of GRIPS 7 (2016-06-21 until 2016-07-18 (27 nights) and 2016-12-11 until 2017-02-01 (50 nights)). As these gaps cover the seasons of annual minimum and maximum temperatures (see also Figure 2, "FoV 3"), they impeded the quality of the fit of the seasonal cycles to such a degree that we refrained from discussing it any further.

4.  Figure 5: Whether GRIPS 8 and GRIPS 7 data separately will show AO and SAO? Since, if AO and SAO are present in the rotational temperature data, it should be visible in individual measurement as well so in OH intensities (e.g., Singh and Pallamraju, Ann. Geophys., 2017, doi:10.5194/angeo-35-227-2017).

    Yes, these oscillations show up in the individual data as well and they are usually the dominant oscillations (see Figure 12 for the AO and SAO in the combined data set). In an earlier study we derived the individual AO amplitudes for the instruments. They don't differ significantly from the values shown in Figure 10. The only complete year with **separate fields of view** for both instruments was **2012**. Then, the annual amplitude of GRIPS 7 was **16.3 K** and the one of GRIPS 8 was **17.3 K**. The 2012 GRIPS 8 value corresponds more or less to the gray curve in Figure 10a (combined data before removal of gaps: 17.2 K). Together with additional data from other sites (OPN, 48.1°N, see Schmidt et al. (2013), AO 2012: **18.1 K**) we assume that the AO amplitude changes with approximately 1K per 100km in latitude across this region of Central Europe.

    AO and SAO of the OH intensities differ significantly from the rotational temperatures, with the amplitude of the intensity SAO being significantly larger than the temperature SAO. Figure R1.4 shows the first, preliminary, assessment of absolute radiances at UFS. In addition to the annual maximum intensities from Nov-Jan, the SAO introduces a secondary maximum in summer from May to July. The explanation of this behavior requires a lengthy discussion of OH temperatures, emission height and atomic oxygen mixing ratio over the course of the year (see e.g., Grygalashvyly et al (2021), DOI: 10.5194/angeo-39-255-2021).

    The results between the instruments only differ significantly whenever the FoV of GRIPS 7 is changed. Figures 10 and 12 cover the behavior of AO and SAO well. However, a detailed discussion of OH intensities is beyond the scope of this study. Thus, no changes to the manuscript have been made.

[Figure]

*Figure R1.4: first (preliminary) assessment of the average seasonal cycle of OH radiances (still without uncertainties, absolute values are subject to change). The data clearly show the presence of an annual oscillation (AO) with a winter maximum from Nov-Jan and a semi-annual oscillation (SAO) causing a secondary maximum from May-Jul.*

**Minor comments:**

1. Page1 line 22: How the precise value depends on details of the analysis?

   Before the analysis we were skeptical whether or not a time lag between solar forcing and OH temperatures exists at all. But Figure 14 shows that such a time lag can improve the correlation considerably. However, the exact value is difficult to determine as is shown in Figure 13a) showing the $R^2$ values for lags of 0-180 days for different parts of the year. While Figure 13 is based on annual means, the result of time-lagged cross-correlation of the smoothed data sets shows maximum $R^2$ values for time lags of 24 to 85 days. Thus, one analysis gives $R^2 > 0.9$ for lags in the range of approximately $105\pm50$ days, the other $R^2 > 0.78$ for $\sim55\pm30$ days; from combining these results we get the stated $90\pm65$ days.

   As was pointed by another reviewer before, this uncertainty of the lag might be due to a residual influence of the 27d variation of the solar forcing. We tentatively assume that the time lag is introduced by a dynamical feedback of the atmosphere to the changed forcing (tides probably come into play here, see e.g. the recent discussion by Gu et al. (2023), DOI 10.5194/egusphere-2023-910 on a very similar topic). The strength of this dynamical forcing will probably vary due to other factors. Thus, the net time lag of the OH temperatures reacting to the solar forcing may actually not be precisely defined but rather depend on local and seasonal characteristics of prevailing atmospheric dynamics.

2. Page 2 line 55-56: sentence may be modified likeà If the instrument cannot be repaired it can be replaced with a similar new instrument.

   changed as suggested, p.2, l. 55-56: If it cannot be repaired the instrument  can be replaced with a similar new instrument.

3. Page 5 line 104 to resolve the P1-linesà to resolve the first three P1-lines (as the first three P1 branch lines are used for the rotational temperature determination, so contamination from nearby P2 branch should not be there).

   Indeed. As can be seen in the right panel of Figure R1.3 there is always one $P_2$-line between two $P_1$-lines of interest, so we rephrased it more clearly, p.5, l. 104-106: The GRound-based Infrared P-branch Spectrometers (GRIPS) are designed to separate  the $P_1$-lines from the neighboring $P_2$-lines of the OH(3-1)-rotational vibrational transition for the derivation of the respective rotational temperature from the $P_1$-lines (that is: a resolving power of $\lambda/\Delta\lambda$ of $\sim500$ or a full width at half maximum (FWHM) of 3.1 nm at 1550 nm).

4. Page 6 line 116: I am just curious to know why GRIPS instruments have been operated with an oblique FoV? Since different FoV will see different part of the sky which needs to be taken care in the data analysis and interpretation of the results. What are the challenges in operating GRIPS in vertical direction in future?

   We operate some GRIPS in vertical direction. It is correct, that some analyses are difficult to perform with an oblique FoV (for example the analyses discussed in Schmidt et al. (2018), DOI: https://doi.org/10.1016/j.jastp.2018.03.002). The GRIPS 7 FoV was changed to study the impact this would have on the data. One part of the respective results is obviously discussed in the current manuscript, other results are discussed by Wüst et al. (2016).
   One big issue at UFS are the environmental conditions. In winter there can be substantial

snow coverage on the roof of up to eleven Meters (!) at certain parts of the building. This even required a static re-evaluation of the building a few years ago. While it might be possible to construct a dome or roof window that can withstand the pressure, the removal of the snow can take days: UFS has more than 480m² of terraces and a roof area comparable in size. In addition, working hours are limited to Monday-Friday, which will then result in substantial data gaps (see https://schneefernerhaus.de/en for further details of the research station).

Another issue is the fact that acrylic domes (typically used to look in the vertical direction) tend to age quickly. This requires frequent recalibration of their transmission. Thus, they are not the first choice for a site with a high amount of UV radiation (see also our answer to question 1.2 "long-term stability" and Figure R1.2).

5. Page 6 line 125: How a minor misalignment affects data cadence?
   The minor misalignment of GRIPS 5 caused a severe reduction of the radiation received by the sensor (i.e. the spectrum wasn't correctly focused onto the photodiode array anymore). In order to preserve a reasonable signal-to-noise ratio the exposure time was doubled.

6. Page 9 line 223: observed volumina à observed volumes
   corrected

7. Page 9 line 224; Wüst et al (2016) à Is it Wüst et al. (2017)?
   We made a mistake here, it is Wüst et al. (2016), but it was accidentally listed as "2015" in the references, since the DOI was already assigned in Dec 2015, but the official publication date was February 2016.

   p. 48, l. 1075-1077, we corrected:
   Wüst, S., Wendt, V., Schmidt, C., Lichtenstern, S., Bittner, M., Yee, J.H., Mlynczak, M.G. and Russell III, J.M.: Derivation of gravity wave potential energy density from NDMC measurements. J. Atmos. Sol.-Terr. Phys. 138-139, 32-46, doi:10.1016/j.jastp.2015.12.003, 2015 2016.

8. Page 11 line 243: while keeping the 180° (tward) azimuth angle. Subset a) à while keeping the azimuth angle of 180°.
   p. 11, l. 243-244, corrected: … not overlap with the FoV of GRIPS 8, while keeping the azimuth angle of 180° (southward) azimuth angle.

9. Figure 4: It is difficult to see dashed green triangle.
   Recognizing the green triangles is indeed difficult to a certain extent. But we are afraid this is mainly due to the properties of the MSIS data. The climatological differences between the individual sites are small and exhibit little variation. Thus, the plot symbols for the 365 values overlap and are almost arranged along a straight line. We tested different plot symbols, sizes and colors, which resulted in virtually no changes. Figure R1.5 shows the differences between putting the MSIS values in the background as was done in the manuscript (left panel) or displaying them in the foreground (right panel). While the MSIS data can be recognized more clearly, they now shade the actual observational data more than we are willing to accept in the context of this study, which focuses on the data.

[Figure]

*Figure R1.5: comparison of the scatter plot with either the green triangles (representing the MSIS climatological values) in the background (left panel) or the black observational data in the background (right panel).*

We added a respective statement in the Figure caption, p. 10, l. 231-233: …processing scheme (see also Figure 6). Note: the green triangles in a) and b) are rather close to each other and appear as one thick green line. Details are discussed in the text.

Note: we also corrected a minor error of the axis label in Figure 4c) (missing "200" between "180" and "220")

10. Page 11 line 43: Subset a) of Fig. 4 à 4a) here and elsewhere. e.g., Page 22 L 247: Subset b) à Fig. 4b)
    removed all occurences of "subset" in the manuscript

11. Figure 5 caption: annual and semi-annual oscillation à annual and semi-annual oscillations; Harmonic Analysis à harmonic analysis
    corrected

12. Page 15 line 348: Sorry for my ignorance but how authors have arrived at the true value of ΔT is often closer to ±2 K (or ±2.75 K/√2)?
    The estimate is based on (statistical) error propagation. Figure 6 displays the temperature difference between the two instruments ($f=T_{G8}-T_{G7}$), most of the results range from -1 K to +4.5 K (=5.5 K or ±2.75 K). Gaussian error propagation applied to f is $\Delta f=\sqrt{((\Delta T_{G7})^2 + \Delta(T_{G8})^2)} \approx 2.75K$. Under the assumption that $\Delta T_{G7}$ equals $\Delta T_{G8}$ this means $(\sqrt{2})* \Delta T_{G7} \approx 2.75K$, which in turn means $\Delta T_{G7} = \Delta T_{G8} \approx ±2K$.
    With different assumptions for $\Delta f \approx 2.75K$ one might arrive at slightly different quantitative estimates. But the main point remains: from the large scatter displayed in Figure 6 follows that $\Delta T_{G7} = \Delta T_{G8} > ± 1K$. Thus, the previous estimate of ±1K acquired from calibrations under laboratory conditions underestimates the true uncertainty of the atmospheric observations.

13. Section 3.4 and Figure11: Replace quasi-biannual with quasi-biennial wherever it appeared in the text.
    replaced at all instances

---

## Referee Report (RR1)

**Manuscript number: AMT-2023-61**

OH airglow observations with two identical spectrometers: benefits of increased data homogeneity in the identification of the 11-year solar cycle-, QBO-induced and other variations

by Carsten Schmidt, Lisa Küchelbacher, Sabine Wüst, and Michael Bittner

I am satisfied with the authors detailed response to my questions and subsequent modifications made in the current form of the manuscript. I therefore find the manuscript suitable for publication in Atmospheric Measurement Techniques in its present form.